# CluMo: Cluster-based Modality Fusion Prompt for Continual Learning in Visual Question Answering

## Abstract

Large vision-language models (VLMs) have shown significant performance boost in various application domains. However, adopting them to deal with several sequentially encountered tasks has been challenging because finetuning a VLM on a task normally leads to reducing its generalization power and the capacity of learning new tasks as well as causing catastrophic forgetting on previously learned tasks. Enabling using VLMs in multimodal continual learning (CL) settings can help to address such scenarios. To improve generalization capacity and prevent catastrophic forgetting, we propose a novel prompt-based CL method for VLMs, namely **Clu**ster-based **Mo**dality Fusion Prompt (**CluMo**). We design a novel **Key-Key-Prompt** pair, where each prompt is associated with a visual prompt key and a textual prompt key. We adopt a two-stage training strategy. During the first stage, the single-modal keys are trained via $K$-means clustering algorithm to help select the best semantically matched prompt. During the second stage, the prompt keys are frozen, the selected prompt is attached to the input for training the VLM in the CL scenario. Experiments on two benchmarks demonstrate that our method achieves SOTA performance.

## 1 Introduction

Visual Question Answering (VQA) is a complicated task, where the goal is to answer questions described in natural language (text) about a given input image. Addressing VQA requires understanding and fusion of information from both the visual and textual domains to generate accurate responses. Recently, significant advancements in addressing VQA tasks have emerged due to the development of pre-trained large vision-language models (VLMs) (Radford et al., 2021b; Kim et al., 2021). However, in real world dynamic scenario (Li & Hoiem, 2017), as data comes in temporal sequence, the VQA model should generate answers based on new images and new questions which has different distribution than original data (Qian et al., 2023a), which lead to the setting of continual learning (CL). In a CL setting, we learn new tasks and aim for continuously improving the model performance without forgetting previously learned knowledge, also known as catastrophic forgetting (French, 1999). To address catastrophic forgetting, a group of CL algorithms are deployed. Regularization-based methods (Kirkpatrick et al., 2017; Li & Hoiem, 2017) constrain the drastic parameter shift when learning new tasks. Expansion-based methods (Douillard et al., 2022; Cai et al., 2023) expand the model with small portion of additional weights and use the expanded weights to learn the new incoming tasks. Rehearsal-based methods (Rebuffi et al., 2017; Rolnick et al., 2019) store a representative subset of the training dataset for each task into a small memory buffer and replay them back during the learning of the current task to maintain the encoded knowledge of the previously learned tasks. More recently, prompt-based methods (Wang et al., 2022b;a) aim to use prompts that contains task-specific or semantic-specific information to prevent catastrophic forgetting. A prompt is attached to the embedded features of the input to adapt the model to focus on the specific characteristics of the input task that has been learned before.

Most existing CL methods consider unimodal, i.e., vision-only or language-only, settings and hence are sub-optimal to address VQA tasks. Applying uni-modal methods does not take the multimodal nature of VQA tasks and leads to limited view. Meanwhile, the uni-modal methods may ignore the rich and complex interaction between modalities and cause deteriorating performance (Qian et al., 2023a). On the other

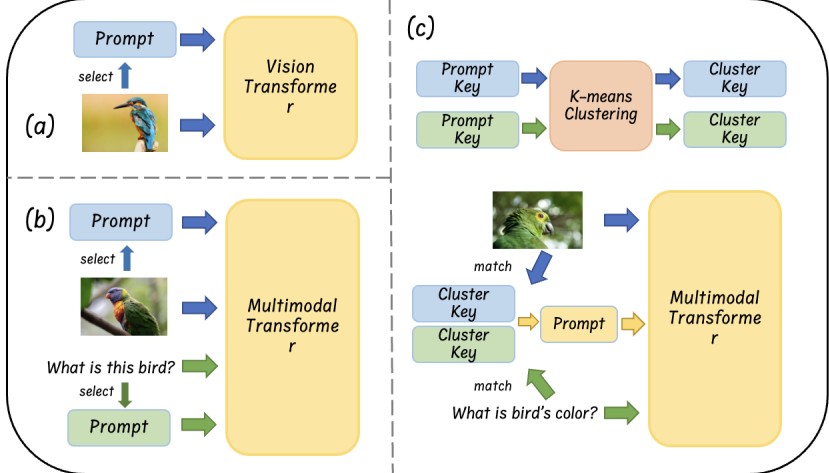

Figure 1: Comparison between existing prompt-based CL methods and our proposed method: **(a)** Uni-modal based methods use image feature to select prompts from a prompt pool. **(b)** Multi-modal based methods use the image features to select image prompts and use the text features to select the text prompts. **(c)** We first train the prompt key using a clustering algorithm to form a cluster key and use the combination of the cluster key from both modalities to select the fusion prompt.

hand, few CL methods are introduced to handle multimodal tasks. Lao et al. (2023); Nikandrou et al. (2024b) adopt knowledge distillation for CL-VQA, and Qian et al. (2023b); Zhang et al. (2023); Lei et al. (2022) adopt prompt-style methods as solution. However, none of them explicitly focus on the interaction between different modalities to boost VQA tasks. To tackle this shortcoming, we propose a novel two-stage prompt learning-based CL method, namely cluster-based modality fusion prompt (CluMo), with the focus of modality interaction. Figure 1 visualizes the high-level idea of our approach. Our method adopts a pre-trained VLM as its backbone and benefits from a clustering-based modal-specific key strategy to boost generalization capacity and minimize catastrophic forgetting. More specifically, we use a clustering-based algorithm to train visual-prompt keys and textual-prompt keys during the first stage. During the second stage, we assign each input image-question pair with well-trained prompt keys to its corresponding visual keys and textual keys. We then use the combination of two modal-specific keys to find the best-matched prompt to adapt the model for the input task. We also benefit from knowledge distillation during training to further improve the performance. Our proposed method outperforms existing alternative methods. Our specific contribution includes:

- We propose a novel clustering-based prompt learning method for training VLMs in CL settings to address VQA tasks with vision-and-language inputs.

- We use a two-stage training strategy to train the prompt keys before training the whole model to guarantee optimal prompt selection that is necessary for generalization on the input.

- We offer extensive experiments to demonstrate that the proposed approach achieves SOTA performance against CL existing methods and offer insight about the reason of this improved performance.

## 2 Related Works

**Visual Question Answering** Visual Question Answering (VQA) has been a pivotal task at the intersection of computer vision and natural language processing which led to advances on more complex tasks. Initially, VQA was formulated as a classification task in which answers are selected from a predefined set of answers (Agrawal et al., 2016) and was solved by using CNNs for image feature extraction and RNNs for

text processing. With the development of transformer and BERT-like models (Lu et al., 2019a; Li et al., 2019), performance in VQA tasks has significantly been improved due to the better capacity of capturing the intricate relationship between two modalities and generating the response in the form of meaningful and descriptive texts. In recent years, VQA task has become downstream task for pretrained large vision language model (VLM) (Liu et al., 2023; Radford et al., 2021a; Li et al., 2021; Lu et al., 2019b), which are initially pretrained on large-scale image-text dataset that are not necessary related to visual question answering. The pre-trained VLMs are essentially capable for zero-shot VQA, and can also be fine-tuned by VQA dataset for VQA downstream task specifically.

**Prompt-Based Learning**   Prompt learning, firstly introduced in NLP tasks (Petroni et al., 2019), is based on providing a fixed function to condition a pre-trained model so that it gets extra information token which specializes it to perform the down-stream task. It is more memory-efficient than using Adapters (Pfeiffer et al., 2021) or LoRA (Hu et al., 2021) and has been used successfully to guide responses of VLMs for a particular task. In general, prompts are not only considered fixed texts, eg. "A photo of", but also can be trainable soft-vectors for both image and text modalities (Lester et al., 2021; Li & Liang, 2021). Zhou et al. (2022) introduces learnable text context, composed of trainable vectors, to replace the fixed textual input. Wang et al. (2022b) adopts trainable vectors append to image input to add additional knowledge to visual input. Our method adopts trainable vector as prompt based on both image and text input.

**Prompt Learning for Continual Learning**   Prompt learning has been used in CL to prevent catastrophic forgetting when a large pre-trained models is trained on a stream of sequentially encountered tasks. It allows a single model to quickly adapt to new tasks in the stream without needing extensive retraining. It allows scalability to learning a large number of tasks since each task primarily requires learning or generating new prompts rather than retraining the entire model. L2P (Wang et al., 2022b) pioneered to connect prompt-based learning and CL. Instead of having a single shared prompt to learn all tasks, L2P introduced the concept of "prompt pool" to maintain prompts for different tasks independently from each other. DualPrompt (Wang et al., 2022a) extended the idea of prompt pool in l2p by introducing task-specific prompt and task-agnostic prompt. S-Prompt (Wang et al., 2023) applied clustering to build the prompt pool with domain-specific prompts. These prompt learning methods for CL only consider single-modality, i.e., vision-only or text-only, and hence are sub-optimal for tasks with multi-modal inputs such VQA when the modalities are related. Our method benefits from the specific properties of multi-modal data to address VQA in CL settings using prompt learning and leads to performance improvements against these methods.

## 3   Problem Description

Consider a set of VQA tasks, $\{\mathcal{T}_i\}_{i=1}^T$, which are encountered sequentially and each of them is from different domain. For each of the tasks, a labeled training dataset $\mathcal{D}^i = \{\langle (\boldsymbol{I}_i^j, \boldsymbol{T}_i^j)^i, y_i^j \rangle_{j=1}^{N_i}\}$ is accessible, $N_i$ denotes the size of dataset, $\boldsymbol{I}_i^j \in \mathbb{R}^{H \times W \times C}$ denotes the input image, $\boldsymbol{T}_i^j \in \mathbb{R}^{L \times |V|}$ denotes the input text, and $y_i^j$ denotes the text-typed discrete label. The order in which the VQA tasks are observed is not known in advance and the training data points are assumed to be drawn iid from a task-specific joint distribution $p_i^t(\cdot, \cdot, \cdot)$. Upon learning each task, the model moves forward to learn the next task. Since all the previously learned tasks can be encountered at any time during testing in the future, the model should learn new tasks such that its knowledge of previously learned tasks is maintained, i.e., by preventing catastrophic forgetting.

We formulate our problem in a domain-diverse **domain-incremental** learning setting (Van de Ven & Tolias, 2019) which assumes all the tasks are from different domains and the boundaries between them are known during training and inference time. Task identities are not known during inference. We consider that each task can be learnt individually by adapting a pre-trained large multimodal transformer $f_{\theta M}^i(\cdot, \cdot)$ via minimizing a suitable discrimination loss $\mathcal{L}$, e.g., cross entropy. In our approach, all the model parameters, except the final classifier layer $\theta_{cls}$, are frozen during training to preserve the generalizability of the model. We benefit from prompt learning to enable using a single model to learn all tasks. To prevent catastrophic forgetting, a trainable task-specific prompt pool, which contains several task-specific prompts, is attached to the model $f_{\theta M}^i(\cdot, \cdot)$ such that the best-semantically-matched prompt is selected based on image and text inputs for task specialization. The prompt is then pre-pended to the input vectors so that the output is

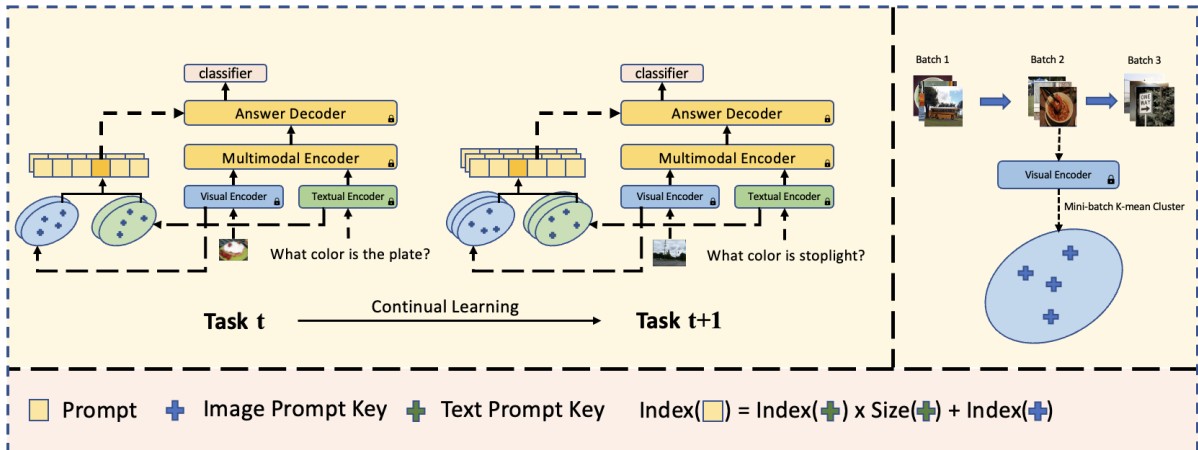

Figure 2: Block diagram of the proposed approach: **Left**: the backbone contains a pre-trained frozen visual encoder, a textual encoder, and a multimodal encoder. The answer decoder shares the same architecture as multimodal encoder. During the training phase, the fixed number of visual prompt keys, textual prompt keys, and a prompt pool will be added for each new task. **Right**: the procedure of visual prompt key training consists of training the modal-specific prompt key by a sequence of randomly selected batches of training data from current task until convergence is reached. Same procedure for textual prompt key.

generated based on specialization. Our method is rehearsal-free and does not need any memory buffer similar to prior approaches (Lopez-Paz & Ranzato, 2022; Rebuffi et al., 2017).

## 4 Proposed Architecture

Our architecture, named **Clu**ster-based **Mo**dality fusion prompt (**CluMo**), contains two unimodal task-specific cluster-based keys for vision and text embeddings and one prompt pool. The combination of the selections from both keys is then used to select the best matched prompt from the prompt pool. A high-level diagram of our approach is presented in Figure 2. In this section, we first introduce the preliminaries such as backbone model and prompt pool-based method in 4.1, and modality fusion prompt in Sec. 4.2, then the cluster-based prompt key is described in Sec. 4.3, and the training and the inference strategy is discussed in Sec. 4.4.

### 4.1 Preliminary

**Backbone**  The base multimodal transformer contains three encoders: the visual encoder $E_v$, the textual encoder $E_t$, and the multimodal fusion encoder $E_f$. Given a image input $\boldsymbol{I}$, i.e., a single image, and a textual input $\boldsymbol{T}$, i.e., a question, the data processing pipeline for the model is:

$$\hat{y}(\boldsymbol{I}, \boldsymbol{T}) = \mathcal{F}(E_f([E_v(\boldsymbol{I}); E_t(\boldsymbol{T})])), \tag{1}$$

where $\mathcal{F}(\cdot)$ is the classifier to predict the answer, and $\hat{y}$ is the predicted answer.

**Prompt Pool**  As an adoption of prompt learning in continual learning, a prompt pool is a set of trainable key-value (**k-p**) pair, in which $\mathbf{k} \in R^{1 \times D}$ denotes the "prompt key", and $\mathbf{p} \in R^{L_p \times D}$ is the prompt. $L_p$ and $D$ denote the length and dimension of the prompt. Given an input image $\boldsymbol{I}$, we compute image feature vector $\mathbf{V}_I = E_v(\boldsymbol{I}) \in R^{L_v \times D}$, where $L_v$ is the dimension of the features, after passing the image through the visual encoder. $\mathbf{V}_I[0]$ is matched with all the keys $\mathbf{k}$ within the prompt pool via similarity score, such as L2 similarity, to find the most similar $\mathbf{k_i}$. In general condition, the corresponding $\mathbf{p_i}$ is selected and prepend to $\boldsymbol{I}$ as $\boldsymbol{I}' = [\mathbf{p_i}; \boldsymbol{I}]$. Parameters of $\mathbf{k}$ and $\mathbf{p}$ are updated through back-propagation during the training. However, in our setting, **the prompt is prepended to the image feature vector $\mathbf{V}'_I = [\mathbf{p_i}; \mathbf{V}_I]$**, and we adopt a two stage training strategy that prompt keys are trained before model and prompts.

### 4.2 Modality Fusion Prompt

Previous prompt-based CL methods such as L2P (Wang et al., 2022b) associate each prompt in the prompt pool with a single prompt key to form Key-Value pair. In practice, the prompt keys in prompt-based CL can be considered as cluster centers. These cluster encode a notion of similarity between the prompts. The input feature vectors that form a cluster in the feature space can be assigned to these cluster centers, which are prompt keys. The intuition behind this idea is that **feature vectors with small geometric distance in the feature space are semantically similar** (Wang et al., 2023).

However, such a key-value pair design considers only single modality without tasks with multimodal inputs. The reason is that different input modalities contain different or complementary semantic information. Hence, having prompt keys that associate with each modality can help guiding prompt selection, which is more comprehensive and representative in term of semantic properties of each modality. Thus, we propose a task-specific prompt pool architecture, namely **Modality Fusion Prompt**, which is composed of the **visual prompt keys $\mathbf{K_v}$**, the **textual prompt keys $\mathbf{K_t}$**, and the **prompt pool $\mathbf{P}$** as following:

$$
\begin{aligned}
\mathbf{K_t} &= [\mathbf{k_{t_1}}, \mathbf{k_{t_2}}, ..., \mathbf{k_{t_{S_t}}}], \\
\mathbf{K_v} &= [\mathbf{k_{v_1}}, \mathbf{k_{v_2}}, ..., \mathbf{k_{v_{S_v}}}], \\
\mathbf{P} &= [\mathbf{p_1}, \mathbf{p_2}, ..., \mathbf{p_{S_p}}], \\
\mathbf{k_{t_i}} &\in R^D, \mathbf{k_{v_j}} \in R^D, \mathbf{p_k} \in R^{L_p \times D},
\end{aligned}
\tag{2}
$$

where $S_t$ , $S_v$, and $S_p$ are the sizes of textual prompt key, the visual prompt key, and the prompt pool, respectively. $L_p$ is the length of each prompt and $D$ is the hidden dimension of the transformer backbone. The prompt pool size $S_p$ is then determined as $S_p = S_v \times S_t$. Each prompt is associated with the unique combination of one visual prompt key and one textual prompt key. Given a specific visual prompt key $K_{v_m}$ and a specific textual prompt key $K_{t_n}$, the Key-Key-Value pair is defined as the following:

$$
(\mathbf{k_{v_i}}, \mathbf{k_{t_j}}) \rightarrow \mathbf{p_{i*S_v+j}}
\tag{3}
$$

As modality fusion prompt is **task-specific**, new visual prompt keys, textual prompt keys and a prompt pool will be initialized for each of the new coming task. The previous ones are frozen during training. Based on our experiment setting, the backbone contains 290.3M parameters, while the modality fusion prompt only contains 73728 parameters for every task, which is only 0.025% the size of original backbone model.

### 4.3 Cluster-based Prompt Key

$K$-means clustering has been widely adopted in machine learning algorithms for semantic separation and understanding, where data from different domains can be explicitly separated via clustering in an unsupervised way (Wang et al., 2023) (Cohn & Holm, 2021). However, even though the data from single task belong to the same domain, they can still be further divided into sub-domains based on the semantic property. To make each prompt key be the semantically cluster center of the sub-domains for both vision and text inputs, we adopt mini-batch $K$-means clustering algorithm on prompt keys of $\mathbf{K_v}$ and $\mathbf{K_t}$ to make each prompt key diverse and representative. Let $\mathcal{B} = (\boldsymbol{I}, \boldsymbol{T})$ be the random batch from the training dataset. We extract the image feature vector $v_I$ and the text feature vector $v_T$ as follows:

$$
\mathbf{V}_I = E_v(\boldsymbol{I}), \mathbf{V}_T = E_t(\boldsymbol{T}),
\tag{4}
$$

where $\mathbf{V}_I \in R^{B \times L_I \times D}$ and $\mathbf{V}_T \in R^{B \times L_T \times D}$, $B$ is the batch size, $L_I$ and $L_T$ are the length of vectors for image and text features, represent the embedded image and text input respectively. For visual prompt key clustering, each image feature vector, $\boldsymbol{V}_{I_n}$, is set by taking mean along second the dimension such that $\hat{\boldsymbol{V}}_{I_n} \in R^{B \times D}$, and $\hat{\boldsymbol{V}}_{I_n}$ is used to compare with every prompt key in $\mathbf{K_v}$:

$$
similarity(n, m) = ||\hat{\mathbf{V}}_{I_n} - \mathbf{K_{v_m}}||_2,
\tag{5}
$$

and the prompt key with highest similarity is assigned to match $\mathbf{V}_{I_n}$. After calculation of the whole batch $\mathcal{B}$, the prompt keys are then updated by calculating the mean of all $\hat{\mathbf{V}}_{I_n}$ assigned to that specific visual prompt key. We repeat the above step until the convergence. The procedure of updating the text prompt key $\mathbf{K_t}$ is similar to updating the image prompt keys. Algorithm 1 summarizes our approach for prompt key training.

---

**Algorithm 1** PromptKeyTraining

---

**Require:** Dataset $\mathcal{D}$, Image Prompt Key Pool $\mathbf{K_v}$, Text Prompt Key Pool $\mathbf{K_t}$, Image Prompt Size $S_i$, Text Prompt Size $S_t$

  **while** Not Converge **do**

    Random Select batch of image $\boldsymbol{I}$, text $\boldsymbol{T}$ from $\mathcal{D}$

    $\hat{\mathbf{V}}_{I_m} = mean(E_v(\boldsymbol{I}), dim = 1)$

    $\hat{\mathbf{V}}_{T_m} = mean(E_t(\boldsymbol{T}), dim = 1)$

    $Cluster_I = $ dictionary()

    $Cluster_T = $ dictionary()

    **for** $i$, $t$ in $\hat{\mathbf{V}}_{I_m}, \hat{\mathbf{V}}_{T_m}$ **do**

      $Key_{img} = $ image key with top $similarity(i, \mathbf{K_v})$

      $Key_{txt} = $ text key with top $similarity(t, \mathbf{K_t})$

      $Cluster_I[Key_{img}]$.append($i$)

      $Cluster_T[Key_{txt}]$.append($t$)

    **end for**

    **for** i in $S_i$ **do**

      $\mathbf{K_v}[i] = mean(Cluster_I[\text{i}])$

    **end for**

    **for** i in $S_t$ **do**

      $\mathbf{K_t}[i] = mean(Cluster_T[\text{i}])$

    **end for**

  **end while**

---

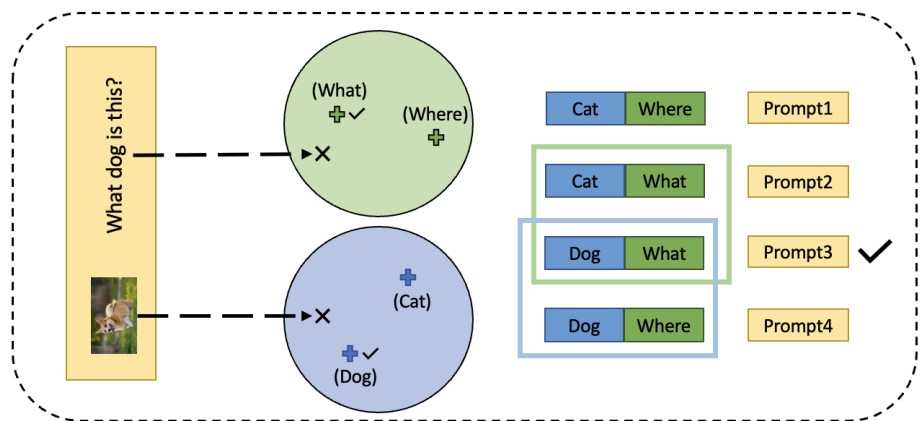

Figure 3: Naive Example of Prompt Selection. Consider a naive animal VQA dataset which only contains dog and cat images and questions only about "what" and "where". During the first stage training, the visual prompt keys and textual prompt keys are learnt to represent "dog"/"cat" and "where"/"what" respectively (in realistic settings the keys will learn the implicit cluster instead of explicit category). Given a test image-question pair, the image and question are projected to their modality-specific feature space through the encoders. The nearest prompt keys, which are keys represent "dog" and "what" are selected. The combination of the two prompt keys lead to "prompt3" which is finally selected.

## 4.4 Training and Inference

---

**Algorithm 2** Continual Learning Procedure

---

**Require:** Datasets **D**, Pretrained Model **M**
  Freeze **M** except **M**.classifier
  **M**.CluMoList = dict()
  **for** dataset $D_i$ in **D do**
    Initialize new CluMo $\mathbf{C}_i$
    **M**.CluMoList$[D_i] = \mathbf{C}_i$
    **PromptKeyTraining**$(D_i, \mathbf{C}_i)$
    Freeze visual key and textual key of $\mathbf{C}_i$
    **for** Batch **B** in $D_i$ **do**
      **for** img $i$,txt $t$ in **B do**
        Select $Key_{img}$ based on $i$ from $\mathbf{C}_i$
        Select $Key_{txt}$ based on $t$ from $\mathbf{C}_i$
        Find prompt $P$ by $Key_{img}$ and $Key_{txt}$ from $\mathbf{C}_i$
      **end for**
      loss = **Train**(**M**,**B**,$P$)
      loss.backward()
    **end for**
  **end for**

---

During training, we adopt a **two-stage training** strategy to ensure that the prompt keys are correctly settled before learning the current task. In the first stage of learning each task $T_i$, we random select batches from the current task's dataloader to train minibatch $k$-means Cluster on the visual and the textual prompt key $\mathbf{K_v}$ and $\mathbf{K_t}$ until reaching the convergence of the clustering algorithm. During the second stage, the trained $\mathbf{K_v}$ and $\mathbf{K_t}$ are frozen. Within the iteration of training dataloader, each training instance is assigned to its nearest prompt key using $k$-nearest neighbor (KNN) algorithm to find the best match prompt $\mathbf{p}_k$ from the prompt pool $\mathbf{P}$. $\mathbf{p}_k$ is then attached to the model pipeline. Inspired by Wang et al. (2022a), we also add a shared general prompt, $\mathbf{p}_g$, with has the same dimension of $\mathbf{p}_k$ for maintaining cross-task knowledge:

$$\hat{y}(\boldsymbol{I}, \boldsymbol{T}) = \mathcal{F}(E_f([\mathbf{p}_g; \mathbf{p}_k; E_v(\boldsymbol{I}); E_t(\boldsymbol{T})])) \tag{6}$$

During the second stage, we also use knowledge distillation to further boost the performance. Before the training of task $T$, we keep a frozen copy of model after finishing $T-1$, denoted as $\mathcal{M}_{T-1}$. To prevent significant parameter shift, we pass the same input image and question to both $\mathcal{M}_T$ and $\mathcal{M}_{T-1}$ and add the difference between the two model's output to the loss:

$$\mathcal{L}_{KD}(\boldsymbol{I}, \boldsymbol{T}) = MSE(\hat{y}_{\mathcal{M}_T}(\boldsymbol{I}, \boldsymbol{T}), \hat{y}_{\mathcal{M}_{T-1}}(\boldsymbol{I}, \boldsymbol{T})). \tag{7}$$

The final objective loss function would be:

$$\mathcal{L} = \mathcal{L}_{ce}(\hat{y}(\boldsymbol{I}, \boldsymbol{T}), y) + \mathcal{L}_{KD} \tag{8}$$

Where $\mathcal{L}_{ce}$ is the same cross entropy loss. The overall training procedure for all tasks come in sequence is presented in Algorithm 2.

During **inference**, the model is frozen and we follow a task-agnostic procedure to let the test instance select the best matched prompt across all prompt pools. The text image-text pair is sent to all the task-specific prompt pool and the image input is aligned with the best-matched image prompt key while the text input is aligned with the best-matched text prompt key within each pool. After getting the best-matched keys, we select the pair of image/text prompt key with the top sum of text similarity and image similarity. The combination of prompt keys is deployed to find the corresponding prompt, which is pre-pend to the output of multimodal encoder. To help better understand the procedure of prompt selection, an visualization example is provided in Figure 3.

| Method | CLOVE-scene | | | | | | | | | | | |
|---|---|---|---|---|---|---|---|---|---|---|---|---|
| | abcdef | | dbafec | | bdcafe | | acbefd | | caefdb | | bafedc | |
| | A ↑ | F ↓ | A ↑ | F ↓ | A ↑ | F ↓ | A ↑ | F ↓ | A ↑ | F ↓ | A ↑ | F ↓ |
| Finetune | 34.03 | 34.28 | 34.89 | 34.99 | 38.83 | 21.65 | 34.45 | 35.14 | 34.42 | 34.47 | 33.95 | 35.67 |
| EWC | 37.49 | 28.04 | 37.00 | 29.10 | 37.95 | 27.46 | 37.99 | 29.68 | 37.13 | 28.63 | 37.83 | 27.97 |
| LwF | 38.18 | 26.82 | 35.03 | 32.84 | 37.31 | 29.11 | 37.85 | 29.87 | 37.94 | 28.15 | 38.21 | 27.48 |
| ER | 41.05 | 19.92 | 42.09 | 17.12 | 42.37 | 18.09 | 41.91 | 20.28 | 41.11 | 19.65 | 42.08 | 20.52 |
| GEM | 41.52 | 18.33 | 43.14 | 14.73 | 42.89 | 17.43 | 42.54 | 20.13 | 41.90 | 20.88 | 43.11 | 19.86 |
| MAFED | 42.72 | 17.23 | 43.48 | 16.53 | 44.22 | 16.90 | 43.53 | 18.12 | 43.03 | 17.21 | 44.68 | 18.04 |
| VQACL | 43.46 | 16.94 | 44.03 | 14.18 | 44.87 | 16.24 | 44.85 | 17.91 | 43.77 | 16.90 | 45.01 | 16.85 |
| L2P | 43.01 | 18.22 | 45.84 | 15.03 | 44.64 | 17.41 | 45.63 | 14.96 | 44.78 | 17.99 | 46.58 | 14.85 |
| DualPrompt | 45.51 | 15.86 | 46.58 | 13.49 | 45.83 | 16.48 | 46.27 | 15.45 | 46.21 | 15.89 | 47.01 | 13.16 |
| S-Prompt | 45.73 | 14.11 | 45.93 | 14.17 | 46.99 | 14.38 | 46.68 | 14.77 | 47.53 | 12.19 | 44.09 | 22.32 |
| **CluMo** | **48.23** | **9.69** | **47.52** | **9.55** | **48.18** | **9.57** | **48.00** | **8.78** | **47.97** | **11.17** | **47.68** | **11.06** |

| Method | CLOVE-function | | | | | | | | | | | |
|---|---|---|---|---|---|---|---|---|---|---|---|---|
| | soarkl | | rsaolk | | osrlak | | oarlks | | skaolr | | ksoarl | |
| | A ↑ | F ↓ | A ↑ | F ↓ | A ↑ | F ↓ | A ↑ | F ↓ | A ↑ | F ↓ | A ↑ | F ↓ |
| Finetune | 31.55 | 53.76 | 37.34 | 39.64 | 23.34 | 57.32 | 24.09 | 62.79 | 16.34 | 74.82 | 17.50 | 84.46 |
| EWC | 35.70 | 47.92 | 37.82 | 41.55 | 38.92 | 40.48 | 40.74 | 33.89 | 37.53 | 37.22 | 40.85 | 32.32 |
| LwF | 37.18 | 46.86 | 36.81 | 44.12 | 39.21 | 39.81 | 36.81 | 41.29 | 30.49 | 53.11 | 29.17 | 55.84 |
| ER | 42.22 | 32.97 | 39.78 | 38.62 | 41.22 | 35.79 | 37.14 | 33.38 | 33.41 | 48.99 | 38.23 | 38.01 |
| GEM | 44.58 | 30.87 | 41.43 | 29.46 | 40.87 | 32.98 | 39.81 | 28.77 | 36.88 | 39.14 | 40.26 | 31.87 |
| MAFED | 43.48 | 31.01 | 40.62 | 34.12 | 42.70 | 30.17 | 40.23 | 26.41 | 35.15 | 44.39 | 40.08 | 34.74 |
| VQACL | 44.36 | 18.28 | 42.89 | 27.45 | 43.58 | 26.86 | 41.25 | 23.53 | 38.67 | 40.31 | 41.84 | 28.65 |
| L2P | 44.80 | 16.38 | 43.39 | 21.26 | 43.27 | 21.97 | 42.54 | 19.18 | 40.4 | 31.92 | 43.37 | 24.19 |
| DualPrompt | 45.01 | 15.90 | 44.26 | 17.43 | 44.66 | 18.50 | 43.69 | 15.31 | 39.32 | 34.78 | 45.65 | 20.54 |
| S-Prompt | 45.45 | 13.47 | 45.01 | 14.76 | **45.27** | 14.29 | 42.98 | 20.20 | 42.85 | 25.82 | 44.09 | 22.32 |
| **CluMo** | **46.72** | **11.14** | **46.29** | **11.28** | 45.13 | **13.52** | **46.55** | **8.21** | **47.04** | **11.15** | **45.77** | **11.69** |

Table 1: Comparative experimental results: the accuracy and forgetting rate for different task order. For each task sequence, **A** ↑ indicates the accuracy of the method, while **F** ↓ is the forgetting rate of each.

# 5 Experiments

Our implementation code is available as a supplement. Please refer to the code to reproduce the results.

## 5.1 Experiment Setup

**Backbone**  We used the public pre-trained large multimodal transformer, ALBEF Li et al. (2021), as our backbone for VQA task. It consists of an image encoder, a text encoder, a multimodal encoder, which uses cross-attention between the two modalities. Specifically for VQA tasks, an pre-trained answer decoder is append after the multimodal encoder, which has same architecture as multimodal encoder.

**Baselines for comparison**  We use seven methods for comparison. We include algorithms from major CL approaches. We include two regularization-based methods: **EWC** (Kirkpatrick et al., 2017) and **LwF** (Li & Hoiem, 2017), four rehearsal-based methods: **ER** (Rolnick et al., 2019), **GEM** (Su et al., 2021), MAFED (Nikandrou et al., 2024a) and VQACL (Zhang et al., 2023). We also include three SOTA prompt-based continual learning methods, **L2P** (Wang et al., 2022b), **DualPrompt** (Wang et al., 2022a), and **S-Prompt** (Wang et al., 2023). We also include finetuning to demonstrate the positive effect of CL. Following the original setting of each method, we leave the whole backbone model unfrozen for non-prompt-based methods and freeze the whole backbone model for prompt-based methods except for the classifier. To make the fair comparison, we fit all the continual learning methods into our backbone, ALBEF, instead of using the original model proposed in each method. Detailed description of each baseline method is presented in Appendix A.5.

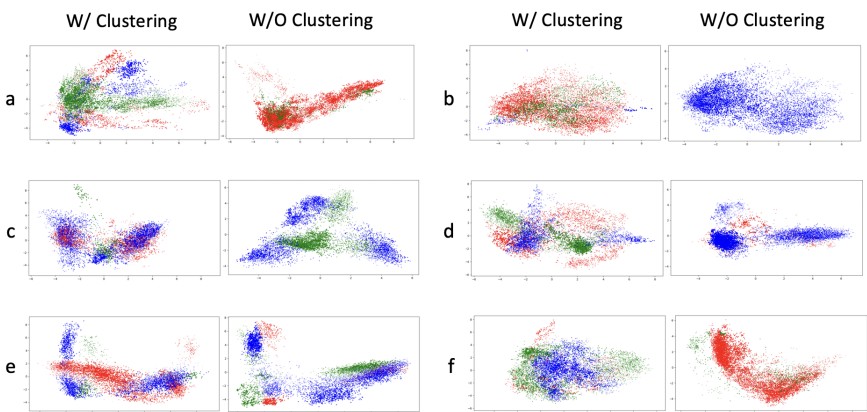

Figure 4: Cluster distribution on all training image data of **CLOVE-Scene**'s six sub-tasks before and after applying mini-batch $k$-means clustering algorithm with visual key size of 3 using PCA. **Color of more diversity indicates more even distribution of key selection.**

**CL Tasks**  We evaluate our method on tasks built using the **CLOVE** (Lei et al., 2022) dataset which is a VQA-based continual learning dataset. The benchmark contains two seperate benchmarks for different scenario, including scene-incremental setting benchmark, **CLOVE-scene**, and function-incremental setting benchmark, **CLOVE-function**. Each of the task sets contains six tasks which are domain-specific and diverse from each other. For more details about **CLOVE** and the tasks we use, please refer to the Appendix A.6 and Table 9.

**CluMo Hyper-parameter**  In all of our comparative experiments, for every task, we set the visual prompt key size and textual prompt key size to be 3, with dimension $1 \times 768$ for each. We set the length of prompt to be 10, hence each prompt has dimension $10 \times 768$. All the choice of hyper-parameters are made by selecting the best-performed ones from multiple analytic experiments.

**Metrics for comparison**  We use the average accuracy and the average forgetting rate on all tasks to evaluate the performance of our method and its ability to tackle catastrophic forgetting. Different from dataset such as **VQAv2** (Goyal et al., 2017b), where each question is paired with different ground truth answers, questions in **CLOVE** dataset only contains exactly one correct answer for each question. Thus, the accuracy is simply calculated by $y == \hat{y}$ for every training and testing data instance. On the other hand, the forgetting rate is calculated as:

$$F = \frac{A_i - A_{ij}}{A_i} \tag{9}$$

where $A_i$ is the accuracy of task $i$, and $A_{ij}$ is the accuracy of task $i$ after the model is trained on task $j$. For details about the optimization and implementation processes, please refer to the Appendix.

## 5.2  Comparative Results

We conduct the comparison experiments on both the **CLOVE-scene** and **CLOVE-function** task sets with a randomly selected task order. In table 1, the task order *abcdef* represents the CL tasks: *ShopAndDining*, *WorkPlace*, *HomeOrHotel*, *Transportation*, *SportAndLeisure*, *Outdoors* in sequence. The *oarlks* in **CLOVE-function** represents tasks: *ObjectRecognition*, *AttributeRecognition*, *RelationReasoning*, *LogicReasoning*, *KnowledgeReasoning* and *SceneTextRecognition*.

We observe in Table 1 that our method outperforms all the baselines across all task order sets in terms of both average accuracy and average forgetting rates. We also observe that the performance of different method within the same group tend to be similar. The regularization-based methods, **EWC** and **LwF**, obtain the sub-optimal accuracy and forgetting rate besides. The reason is that the domain for each task in

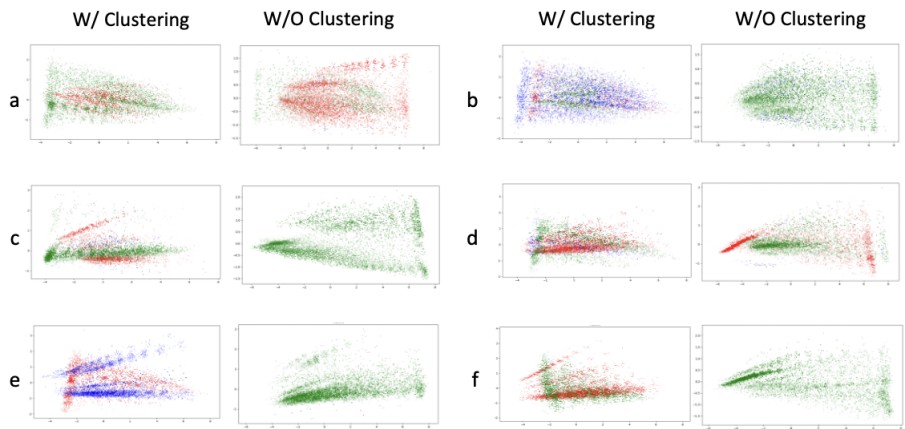

Figure 5: Cluster distribution on all training text data of **CLOVE-Scene**'s six sub-tasks before and after applying mini-batch $k$-means clustering algorithm with textual key size of 3 using PCA.

Table 2: Ablative Experiments

| Methods | Accuracy | Forgetting |
|---|---|---|
| Full Method | 48.23 | 9.69 |
| Ablative KD | 46.67 | 12.97 |
| Ablative Clustering | 47.36 | 10.09 |
| Ablative Textual Key | 47.45 | 10.05 |
| Ablative Visual Key | 47.23 | 10.54 |

the dataset is significantly different from the rest of tasks and hence regularization methods fail to capture the common space of the parameter distribution. This challenge makes it difficult to maintain the accuracy of the current task and previous tasks at the same time using regularization. The replay methods, **ER**, **GEM**, **MAFED** and **VQACL**, achieve better performance than regularization-based methods. This can be explained by the fact that replaying the data from previous task is an efficient way to remind the model and adjust its parameter distribution not too diverse from previous ones. However, because we need to rely on a memory buffer to store samples for replay, these methods are memory-consuming and thus not space-efficient. Moreover, replay-based methods are still limited by the upper-bound of joint training, as they generally can only reduce catastrophic forgetting without boosting the accuracy of individual tasks. On the other hand, the prompt-based methods, namely **L2P**, **DualPrompt**, and **SPrompt**, achieve superior performances compared to more traditional CL methods. Rather than tuning the whole model with regularization, prompt-based methods store the prior knowledge in trainable prompts, which are smaller and more efficient than memory buffer, and keep the main body of backbone model frozen. With the combination of generalization capacity of pre-trained model and specific previous knowledge stored in prompt, prompt-based method can outperform other methods. Among all the methods, our method achieve the best performance.

Compared with the baseline prompt-based methods which only consider visual modality for prompt selecting and updating, **CluMo** takes care of both the visual and textual modalities, as well as the fusion of the two for selecting the prompt which deploys the given information more comprehensively to process the prompt. Our design thus fits better in multimodal learning scenario than other existing continual learning methods.

### 5.3 Ablation Experiments

To offer a better insight about our method, we perform an ablation study for each component of **CluMo** to study the positive contribution of each component. We study the effect of the following:

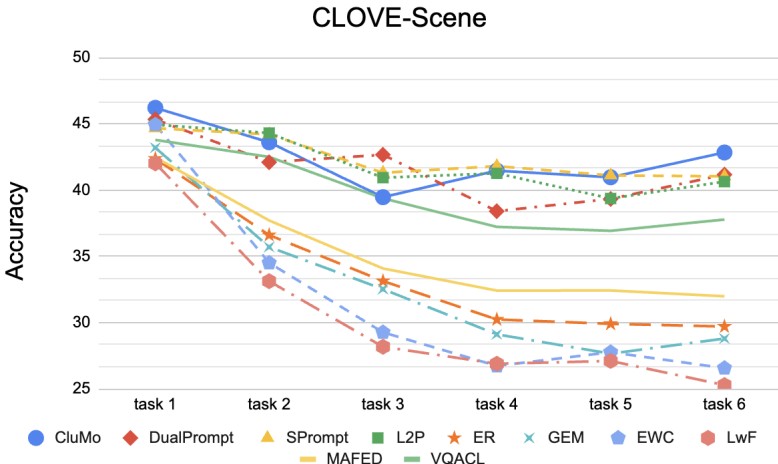

Figure 6: **Accuracy** of the first task after the model trained on the following tasks. The leading performance of **CluMo** indicates its leading capacity of prevent catastrohpic forgetting.

- **Visual Prompt Key**, key to separate the inner-task image features by their semantic property.

- **Textual Prompt Key**, key to separate the inner-task text features by their semantic property.

- **Minibatch $k$-means Clustering** which train the prompt keys as centers of clustering algorithm to better fit the semantic meaning.

- **Knowledge Distillation**, to prevent the drastic parameter shift of unfrozen classifier.

We conduct ablation experiment on **CLOVE-scene** dataset with the task order *abcdef*. We set the size for both the visual prompt key and the textual prompt key to be three. For ablative text experiments, we remove the text prompt keys and change the size of visual prompt keys to 9 to achieve the same total prompt size. We also removed the visual prompt key which is the same for ablative visual experiments.

Results for this experiment is presented in Table 2. We observe that despite having the same number of prompts, the performance values of Ablative Textual Key and Ablative Visual Key are lower than our full pipeline. This result verifies our hypothesis that both modalities should be used to guide the prompt selection and the missing of any will cause information lost and lead to sub-optimal performance. In other words, current approaches for unimodal settings do not use all the information we have in multimodal scenarios. We also observe that without the clustering algorithm, the performance of ablative clustering is the lowest among all the settings which indicate the significance of doing cluster training for learning the prompt keys.

### 5.4 Analytic Experiments

**Effect of clustering** To show the effect of clustering algorithm, we empirically show the correlation between the clustering error and the downstream accuracy. As we apply Euclidean distance as metric to learn the clusters, we record the average distance between each point to its assigned cluster center for every task, and take the average for all the tasks:

$$\mathcal{E} = Avg(\sum_{i=1}^{N} Avg(\sum_{j=1}^{M} ||x_j - c_k||_2)) \tag{10}$$

where $i$ represent the number of tasks, $j$ represent the training data from task $i$ and $k$ is the $k^{th}$ cluster center. We consider both the visual prompt key training and the textual prompt key training in this experiment. Table 4 presents the results. Same as our expectation, we observe a negative correlation between the

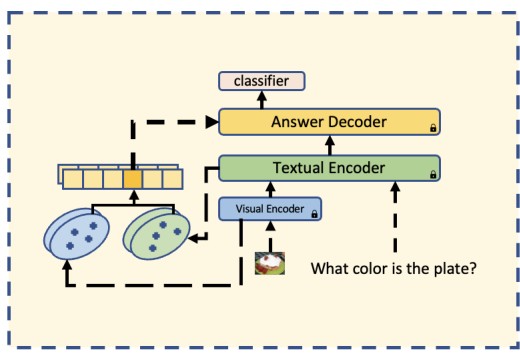

Figure 7: CluMo-BLIP architecture

| Method | CLOVE-scene | | | | | | | |
|---|---|---|---|---|---|---|---|---|
| | abcdef | | dbafec | | acbefd | | caefdb | |
| | A ↑ | F ↓ | A ↑ | F ↓ | A ↑ | F ↓ | A ↑ | F ↓ |
| Finetune | 40.78 | 22.23 | 41.59 | 23.28 | 38.32 | 25.54 | 40.80 | 20.21 |
| EWC | 42.08 | 17.23 | 42.44 | 20.7 | 40.86 | 18.69 | 41.02 | 19.6 |
| LwF | 41.32 | 19.8 | 42.07 | 19.25 | 42.35 | 17.88 | 41.67 | 18.20 |
| ER | 44.43 | 7.90 | 44.72 | 12.84 | 42.82 | 13.15 | 43.87 | 11.83 |
| GEM | 45.34 | 9.78 | 46.01 | 9.39 | 43.11 | 11.49 | 44.23 | 10.04 |
| L2P | 44.37 | 10.67 | 43.15 | 11.69 | 44.64 | 9.58 | 44.31 | 11.04 |
| DualPrompt | 45.91 | 5.27 | 45.72 | 8.08 | 46.00 | 6.92 | 46.35 | 7.88 |
| S-Prompt | 46.94 | 4.19 | 47.36 | 4.08 | 46.12 | 5.34 | 47.17 | 3.52 |
| **CluMo** | **47.60** | **12.32** | **48.62** | **10.98** | **47.86** | **12.67** | **48.85** | **9.13** |

Table 3: Comparison Experiment on CLOVE_scene dataset with **BLIP** as backbone.

clustering error and the performance accuracy, in another word, lower $\mathcal{E}$ for image and text prompt keys leads to a higher accuracy. Without the clustering component, we observe $\mathcal{E}$ to be as high as 42.38 and 42.8 for image prompt key and text prompt key, respectively. After applying clustering algorithm, $\mathcal{E}$ drops below 20, which can be considered significant, for both modalities and the accuracy improves 2.97%.

**Additional Backbone**   In comparative experiments, we adopt **ALBEF** (Li et al., 2021) as our back bone, leveraging its architecture for a consistent foundation across our experiments. While only testing the baselines with a single backbone transformer might be biased and to further show **CluMo**'s generalizablity to other VLMs, we evaluate **CluMo**'s performance on **BLIP**, a different transformer architecture than **ALBEF**, (Li et al., 2022). Without multimodal encoder that takes in both visual and textual feature, architecture such as **BLIP** send visual feature into text encoder and fuse it with cross-attention block. As the architecture is different, we present the adoption of **CluMo** to BLIP in Figure 7. Similar to the comparative experiments in main sections, we choose **CLOVE-scene** dataset as benchmark and randomly select 4 difference task orders to perform experiments. The results are presented in Table 3. As **BLIP** is more recent and SOTA model than **ALBEF**, we observe that the overall performance of **BLIP**'s experiments get higher accuracy than those of **ALBEF**, except for prompt-based methods. During experiments, we found that with **ALBEF**, attaching prompts to input features can naturally improve the accuracy of given tasks. In the case of **BLIP**, attaching the same prompt will harm the accuracy, which makes the performance of **BLIP**'s prompt-based methods similar to **ALBEF**'s results. However, even though we observe a deficiency, in some extent, of prompt-based to **BLIP**, **CluMo** still obtains the best accuracy and forgetting rate compared with all the baselines.

**Cluster Visualization**   To show the effect of clustering on prompt key more intuitively, we visualize the visual prompt key selection distribution and textual prompt key selection distribution on the visual and

textual portion of the training data for **CLOVE-Scene** in Figure 4 and Figure 5. Since we use three visual prompt keys for each task, the vector features of visual data are split into three groups, which are the green, blue and red points in Figure 4. We observe that without using clustering, visual data are more likely to overlap on the same cluster center which means they would lead to select the same visual prompt key. After performing clustering, we observe that the distribution becomes more evenly, and every cluster of data is diverse and separated from the others which means that the visual data can be separated explicitly. The diversity of prompt key selection indicates each input image can find the "correct" prompt which is semantically closer to it, which contains more specific knowledge about the sub-domain the given image belongs to. The visualization of textual prompt keys indicates similar observation.

Table 4: Acc. with different clustering error

| $\mathcal{E}$. Image | $\mathcal{E}$. Text | Accuracy |
|---|---|---|
| 15.40 | 10.72 | 48.23 |
| 15.74 | 12.22 | 47.89 |
| 17.21 | 12.53 | 47.21 |
| 42.38 | 42.8 | 47.06 |

Table 5: Acc. with different prompt key size

| $S_{img} \times S_{txt}$ | Accuracy |
|---|---|
| $1 \times 1$ | 46.01 |
| $2 \times 2$ | 47.49 |
| $3 \times 3$ | 48.23 |
| $4 \times 4$ | 47.71 |
| $5 \times 5$ | 47.45 |
| $10 \times 10$ | 48.12 |

Table 6: Acc. with length/number trade-off

| $S_v \times S_t \times L_p$ | Accuracy |
|---|---|
| $2 \times 2 \times 22$ | 47.82 |
| $3 \times 3 \times 10$ | 48.23 |
| $4 \times 4 \times 6$ | 47.45 |

Table 7: Acc. with different freezing layer

| Freeze Num. | Accuracy |
|---|---|
| 6 | 48.23 |
| 5 | 44.66 |
| 3 | 47.46 |
| 0 | 37.48 |

**Tracking the Accuracy for the First Task**  To take a closer look into the effect on preventing catastrophic forgetting and increasing the accuracy in CL, we track the accuracy of the first task while learning the task sequence. The result is shown in Figure 6. We see that the accuracy drops until task 4, and then slightly increases until task 6. Although it is not our main focus, this behavior shows a trend of forward transfer between the tasks. Among all the baseline methods, we notice that prompt-based methods, **SPrompt**, **DualPrompt** and **L2P**, significantly outperform other methods which verifies the SOTA status of prompt learning in CL and its success in preventing catastrophic forgetting. Our method **CluMo**, on the other hand, still outperform all prompt-based baseline methods. We observe that using the cluster-based prompts, the accuracy on the first task is superior compared to the other methods at the very beginning. Similar to other prompt-based method, our method's accuracy slightly drops until task 4 and improves subsequently. As the accuracy of our proposed method is higher than others at all time steps, our method has the leading performance in terms of both accuracy and backward transfer.

**Effect of Frozen Decoder Layers**  Although previous prompt-based methods keep all the transformer layers frozen during the training, we still conduct experiment on freezing different number of transformer layers to check effect of it. The results are presented in Table 7. We observe keeping whole transformer layers frozen achieve the best performance, and unfreeze all gets the worst. However, freezing 5 layers get worse result than freezing 3 layers. Our hypothesis is that freezing 5 layers makes the model vulnerable to parameter shift yet not flexible enough to learn new knowledge through the trainable transformer layer and only freezing 3 layers makes the model capable for adopting new knowledge with three unfrozen layers. Overall, freezing all the transformer layers achieves the best accuracy and the result matches the choice of freezing model for **L2P**, **DualPrompt** and **SPrompt**.

**Effect of Prompt Key Size**  We also conduct an experiment to study the effect of prompt pool size to show the stability of our method with respect to this hyper-parameter. In Table 5, we choose different visual

prompt key and textual prompt key sizes $1 \times 1$, $2 \times 2$, $3 \times 3$, $4 \times 4$, $5 \times 5$, $10 \times 10$, corresponding to 1, 4, 9, 16, 25, and 100 prompt pool sizes. When we have prompt key size of 1, which indicates every task has no choice but only one prompt, the accuracy drops significantly. Other than prompt size 1, we find that choosing prompt size 3 has the best result, 47.23. We also observe that having larger prompt size than 3 results in decreasing of performance. This observation possibly indicate that dividing a single task into four sub-domains is specific enough for prompts to learn fine-grained sub-domain information, as the key size continues increasing, the more-detailed information the prompts learnt won't boost the performance significantly. Further explanation can be found in Appendix A.4.

**Prompt Length and Prompt Size**  In the experiment setting, we utilize 3 visual keys and 3 textual keys, and each prompt has length of 10. In sum, we have $3 \times 3 \times 10 = 90$ total prompt length for each task. We conduct experiments to show the trade-off of prompt length and prompt numbers given the fixed total prompt length. To make a fair comparison, we keep the visual key and textual key the same size, and choose the closest integer prompt length to let the multiplication of the three be around 90 . Within table **??**, among the three different combination, we find that the current setting, which is the most balanced one, obtains the best result. Regarding the rest of the two, the higher prompt length has the better performance.

# 6 Conclusion

We introduced a novel prompt-based continual learning method for learning multimodal tasks. While most of existing methods apply simple prompts on a single modality, our method proposes modal-specific visual prompt keys and textual prompt keys and train them to capture the semantic properties of the training dataset using K-means clustering algorithm. We use the combination of both the visual prompt key and the textual prompt key to select prompts, which enable the prompt to better boost the performance. Our experiments show that our method achieves the state-of-the-art performance in continual VQA tasks in different domains compared to other regularization-based, rehearsal-based and prompt-based CL methods.

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

| Method | COCOQA-OKVQA-VQAv2-TEXTQA-GQA | | | | | |
| | COVTG | | VGOCT | | OGCTV | |
| | A ↑ | F ↓ | A ↑ | F ↓ | A ↑ | F ↓ |
|---|---|---|---|---|---|---|
| Finetune | 28.11 | 38.41 | 34.84 | 34.27 | 31.02 | 19.97 |
| EWC | 29.94 | 33.02 | 36.00 | 29.10 | 34.41 | 16.72 |
| LwF | 30.72 | 31.93 | 35.03 | 32.84 | 33.65 | 18.08 |
| ER | 32.33 | 24.68 | 36.25 | 27.12 | 35.87 | 15.31 |
| GEM | 34.07 | 20.19 | 37.62 | 19.28 | 36.52 | 14.61 |
| MAFED | 34.65 | 18.68 | 38.22 | 17.41 | 36.23 | 15.15 |
| VQACL | 35.89 | **16.50** | 39.83 | 15.31 | 37.08 | 13.22 |
| L2P | 34.90 | 21.56 | 38.70 | 17.11 | 35.90 | 14.69 |
| DualPrompt | 35.21 | 22.07 | 39.23 | 16.36 | 37.15 | 12.88 |
| S-Prompt | 36.08 | 19.41 | 40.17 | 13.85 | 37.95 | 11.97 |
| **CluMo** | **39.90** | **8.62** | **40.83** | **12.60** | **38.29** | **13.63** |

Table 8: Comparison experiments on dataset consists of **C**:COCQA, **O**:OKVQA, **V**:VQAv2, **T**:TEXTQA and **G**:GQA with different task orders.

# A   Appendix

## A.1   Extra Experiments

To further evaluate **CluMo** compares to other baseline methods, and should the generalizability of **CluMo** to other data, we introduces extra dataset as supplement experiments. We design VQA-continual learning setting with five datasets: **COCOQA** (Ren et al., 2015), **OKVQA** (Marino et al., 2019), **VQAv2** (Goyal et al., 2017a), **TEXTQA** (Singh et al., 2019), **GQA** Ainslie et al. (2023). Due to the computational limit and the unbalance of each task's size , we restrict each training dataset size to 20K, and each testing dataset size to 3K by selecting the top 20K/3K data from the dataset and thus form it in a way similar to the scale of CLOVE dataset. The comparison result is presented in table 8. From the table, we observe that task order has a relatively significant effect on the performance, which, based on our hypothesis, is due to the difference similarity between each task. For example, COCOQA and VQAv2 has similar benchmark design, while TextQA specifically focuses on understanding the text in within the image. When learning similar tasks, the forward transfer is more likely to happen and the forgetting rate will drop. On the other hand, learning dis-similar tasks in sequence will cause forgetting rate to increase. **CLUMO** still beats other baseline methods with different task orders.

## A.2   Hardware Setup and Hyper-parameter

All experiments were conducted using a single Nvidia A40 GPU. We employed the AdamW optimizer across all experiments, utilizing a cosine learning rate scheduler, and set the initial learning rate to $lr = 3 \times 10^{-4}$. The models were trained for 5 epochs with a batch size of 16.

For **EWC**, we set the fisher sample percentage to be 0.1 and ewc loss weight equals to 0.1 as well.

For **Experiment Replay**, we store 1% of data from each tasks into the memory buffer. During the training of the current task, we randomly select a batch of data from the memory buffer to train the model for every 100 batches of current data training.

For **GEM**, we also store 1% of data to memory buffer, which is randomly picked from each tasks.

For **CluMo** framework, we configured the visual prompt key size ($S_v$) and the text prompt key size ($S_t$) both to 3, with the prompt length ($L_p$) set to 10.

For **L2P**, we set the prompt pool size equals to 20 and prompt length to be 5.

For **DualPrompt**, the G-Prompt was inserted into layers 0 and 1 of the visual encoder, while the E-Prompt was integrated into layers 2, 3, and 4.

For **S-Prompt**, we set the prompt length to be 10 and prompt pool size to be 30.

Furthermore, for prompt-based techniques such as L2P, DualPrompt, and SPrompt, we opted to freeze the entire backbone model, allowing only the final classifier layer to remain trainable. Conversely, for all other baseline methods, no parameters were frozen, ensuring the entire network was fine-tuned during training.

### A.3   Time Complexity Analysis

Given that the model is trained on each task for 5 epochs in our experiment setting, the total time to train a task with CluMo is about 1600 seconds, among that the training of the cluster takes about 60 seconds in total. While fine-tuning the ALBEF model on the same task is about 1450 seconds, which is about 10% slower than directly fine-tuning the model.

### A.4   Further Explanation of Prompt Key Size

In main section, we observe that the performance of **CluMo** is not significantly affected by prompt key size. Based on the results, we propose further hypothesis that: each prompt should keep the balance between "general" knowledge and "specific" knowledge to improve the result. Without sub-domain division, the prompt learns mostly the general knowledge across the task. Letting each prompt match a sub-domain of the task can improve the accuracy by focusing on sub-domain-specific knowledge, but further separating the task into smaller sub-domains may harm the learning of "general" knowledge, which prevents the improvement of the model's performance or even harm the performance a little bit.

### A.5   Baseline Methods Description

We provide the detailed description for the function of every baseline method here to present more comprehensive understandings for the comparison experiments.

**EWC**   works by slowing down the changes to the important parameters for previous tasks when training on new tasks. The importance of each parameter is calculated through **Fisher Information Matrix**, the larger the fisher value, the more critical the parameter is. It achieves this by adding a regularization term to the loss function that penalizes large updates to these critical weights.

**LwF**   works by preserving knowledge from previous tasks using a knowledge distillation approach, where "soft targets" from the previous task act as a regularizer when learning new tasks. While training on new task, LwF freezes a copy of model trained on previous tasks, serving as **teacher model**, and apply a "soft prediction" using teacher model as an additional regularizer to prevent forgetting.

**ER**   works by maintaining a memory buffer to store a limited number of samples or intermediate features from previously learned tasks. These stored samples are then reused when learning new tasks with specific frequency. The buffer usually has a fixed size, and new experiences replace older ones once the buffer is full.

**GEM**   works by using a memory buffer, named "episodic memory" to store a subset of examples from previously learned tasks. During training, it computes gradients for both the current task data and the stored examples and ensures that the gradient update for the current task does not increase the loss on the examples from the previous tasks to prevent the catastrophic forgetting.

**L2P**   works by maintaining a shared uni-modal pool of prompts across all the tasks that are used to condition the model when learning new tasks. These prompts act as a kind of "memory" that encodes relevant information from previous tasks and domains. Each task is assigned one or more prompts from the pool, and these prompts are selected based on their relevance to the new task at hand.

**DualPrompt**   works by maintaining two kind of uni-modal prompts, **Expert Prompts** (task-specific prompt), designed to handle knowledge that is unique to specific tasks, and **General Prompts**, designed to handle general knowledge that can be transferred across multiple tasks. During training, the model

Table 9: size of each task in CLOVE

| task | training size | testing size | image source |
|---|---|---|---|
| ShopAndDining | 20K | 3K | MS-COCO |
| WorkPlace | 20K | 3K | MS-COCO |
| HomeOrHotel | 20K | 3K | MS-COCO |
| Transportation | 20K | 3K | MS-COCO |
| SportAndLeisure | 20K | 3K | MS-COCO |
| Outdoors | 20K | 3K | MS-COCO |
| ObjectRecognition | 20K | 3K | MS-COCO |
| AttributeRecognition | 20K | 3K | MS-COCO |
| RelationReasoning | 20K | 3K | MS-COCO |
| LogicReasoning | 20K | 3K | MS-COCO |
| KnowledgeReasoning | 20K | 3K | MS-COCO |
| SceneTextRecognition | 16.8K | 2.4K | VG |

dynamically selects **Expert Prompts** and **General Prompts** to help process new tasks while maintaining performance on previously learned tasks to balance both task performance and overall generalization.

**S-Prompt** works by maintaining an uni-modal prompt pool across different domain of tasks. Within the prompt pool, the prompts from same domain are aligned closer in the feature space with K-mean clustering algorithm. While new task comes in, new set of prompts are added into the prompt and trained to be separated from other prompts via K-mean clustering. During inference stage, the prompts with best alignment with input visual feature is dynamically selected.

### A.6 CLOVE dataset detail description

In the **CLOVE-Scene** and **CLOVE-Function** datasets, all tasks have a uniform distribution of training and testing data, with the exception of the *SceneTextRecognition* task, which comprises 16.8K training samples and 2.4K testing samples. The remaining tasks within these datasets contain 20K training samples and 3K testing samples each. It is reflected in the Table 6.

To provide a deeper understanding of the **CLOVE** dataset, we offer additional details here. We have included visualizations in Figure 8 and Figure 9 to showcase two sample images from each task within the datasets, emphasizing the distinctiveness of each domain. From these samples, it is evident that the images in the **CLOVE-Scene** dataset vary significantly across tasks, even though the questions associated with them are similar in structure, differing primarily based on the content depicted in the images.

On the other hand, the **CLOVE-Function** dataset presents a different scenario. While the images across various tasks may appear to belong to similar or overlapping domains, making it challenging to distinguish one task from another based solely on visual content, the diversity becomes apparent when considering the questions. Each task within the **CLOVE-Function** dataset involves distinct types of reasoning, as reflected in the varied nature of the questions posed, which are tailored to serve different reasoning purposes.

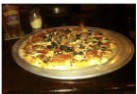 Q: Is there a sandwich on the tray?
A: No.

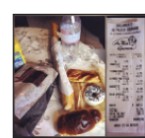 Q: Is the bottle on the couch?
A: Yes.

Shopping and Dining

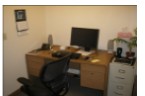 Q: Which room is it?
A: Office.

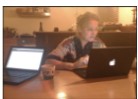 Q: Is a laptop to the left of her?
A: Yes.

Workplace

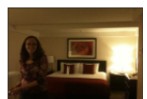 Q: Does the blanket look red?
A: Yes.

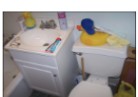 Q: What toy on the top of sink?
A: Rubber Duck.

Hotel

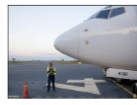 Q: What is the plane behind the man?
A: Runway.

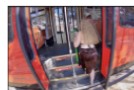 Q: What are the letters on?
A: Stairs.

Transportation

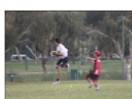 Q: Is the man on the right of frisbee
    wearing a hat?
A: Yes.

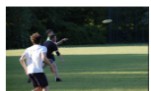 Q: What is the man playing?
A: Frisbee.

SportAndLeisure

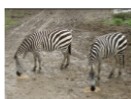 Q: Does the zebra nose have the
    white color?
A: No.

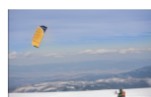 Q: What is flying in the sky?
A: Kite.

Outdoors

Figure 8: CLOVE-scene dataset samples

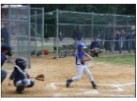 Q: What color is the helmet in the middle of the image?
A: Blue.

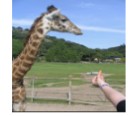 Q: Is it indoor or outdoor
A: Yes.

**Attribute Recognition**

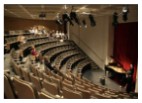 Q: Which red object behind the black piano can keep light out of the room?
A: Curtain.

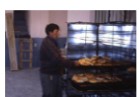 Q: What food is next to the object that I can use to for making toast?
A: Bread.

**Knowledge Reasoning**

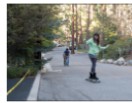 Q: What do the marker and the post have in common?
A: Color.

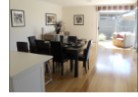 Q: Is the color of pillow different than that of counter?
A: Yes.

**Logic Reasoning**

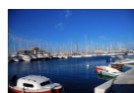 Q: What place is it?
A: Harbor.

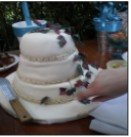 Q: What is this, a couch or a table?
A: Table.

**Object Recognition**

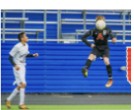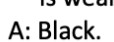 Q: What color is the jersey the boy is wearing?
A: Black.

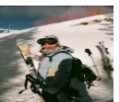 Q: Who is wearing goggles?
A: Woman.

**Relation Reasoning**

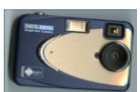 Q: What is the brand of this camera?
A: Dakota.

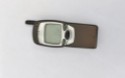 Q: What is the brand of the phone?
A: Nokia.

**Scene Text Recognition**

Figure 9: CLOVE-function dataset samples

