# OpenReview forum: "CluMo: Cluster-based Modality Fusion Prompt for Continual Learning in Visual Question Answering"
_TMLR — Rejected by TMLR_

### Review · Reviewer_Aqr9 · 2024-10-14

**Summary Of Contributions:**

This paper presents a novel approach for multi-modal continual learning (CL), specifically designed to handle tasks involving both vision and language, such as Visual Question Answering (VQA). Existing *CL* methods are typically developed for a single modality and therefore struggle to address multi-modal tasks like *VQA*.

In conventional prompt learning, each data point is mapped to a key, and the corresponding task prompt is selected based on that key. In contrast, the proposed **CluMo** method introduces separate key pools for each modality, selecting prompts based on a unique combination of a text key and a vision key.

**CluMo** is trained in two stages. First, prompt keys are learned using mini-batch K-means clustering. Afterward, the key pools are frozen, and the prompts are trained end-to-end on the *VQA* tasks, while the backbone model remains frozen.

The experimental results demonstrate two key findings: 1) **CluMo** significantly outperforms other *CL* methods, and 2) detailed ablation studies offer insights into **CluMo**'s mechanisms and effectiveness

**Audience:**

Yes

**Broader Impact Concerns:**

There are no broader impact concerns

**Claims And Evidence:**

No

**Requested Changes:**

See weaknesses section.

**Strengths And Weaknesses:**

## Strengths
- This work introduces a new continual learning (CL) method using prompt learning, specifically tailored for vision-language models (VLMs).
- The methodology is well-explained, though the notation could be improved for clarity.


## Weaknesses/Requested changes

- The paper would benefit from clearer writing, as the current notation is somewhat inconsistent. It mixes matrix, vector, and scalar notation, which can lead to confusion. Standardizing the notation throughout would improve readability.

- Please use \citep{} for inline citations when not directly referring to the authors or the specific work.

- Most existing *CL* methods are unimodal, focusing on either vision-only or language-only tasks. However, just because these models are designed for a single modality does not necessarily mean they cannot be applied to *VQA* tasks, as the authors suggest. What specifically prevents these methods from being adapted to a multi-modal setting? It would be helpful if the authors could further explain or provide arguments as to why current methods fail in multi-modal contexts.
Additionally, in Section 5, unimodal *CL* methods are used as a baseline for comparison, which seems somewhat contradictory with the main motivation of this paper.

- All model parameters, except for the final classifier layer, are frozen during training to preserve the model’s generalizability. However, the entire purpose of *CL* is to prevent the model from overfitting to a new task while still allowing it to adapt. Wouldn't it be more appropriate to train the entire model end-to-end to better demonstrate the strength of both the *CL* baselines and **CluMo**?
By freezing the backbone, you effectively prevent any catastrophic forgetting in the backbone model, but by training only the classification layer, the model is forced to capture all task-specific knowledge through a single layer. How well does **CluMo** perform when the entire backbone, or at least several layers, are fine-tuned? This could provide a clearer comparison of its effectiveness.
- By introducing an additional key pool to the learning algorithm, more trainable parameters are effectively added. It’s not surprising that this leads to improved performance. However, what would happen if the authors used only a single-modality key pool (either vision or language), but doubled the number of key prompts? This would help clarify if the improvement comes from the multi-modal approach or simply from increasing the number of parameters.
- Do the other baseline methods also incorporate knowledge distillation? If not, please include an ablation study to show the impact of knowledge distillation on the results.
- **Table 4** presents some puzzling results, as the number of keys appears to have no noticeable effect. Could this be explained more clearly? Additionally, what happens if a 1x1 key configuration is used? This could provide more insight into the role of the key size in the model's performance. Because having the same performance, regarding the number of prompts does not show that the prompts have much effect.
- In my opinion, this work lacks sufficient motivation. Could the authors provide a real-world example to better justify the approach? When would someone need to sequentially train a *VLM* on different tasks, and how does this scenario differ from large-scale *VLM* training? Clarifying these points would help establish the relevance and necessity of this work

---

> ### Author Response · Authors · 2024-11-04
> **Rebuttal**
>
> Dear Reviewer,
>
> Thank you for your time and efforts. We are glad that you have found the leading performance of CluMo and the methodology to be well-explained. We thank you for the suggestion that helped improve our work. We’ve modified the paper content and marked them in blue where we have reflected your concerns. In our experiments, we added a new backbone, a new dataset and two more baselines for further comparison and analysis. We hope you engage in discussion and give us a second chance until we address all your concerns. Our responses to the weaknesses you raised are below:
>
> $\textbf{1. The paper would benefit from clearer writing...}$
>
> Thank you for pointing this out so we can make our writing clearer. We’ve modified the notations and marked them blue in section 4. Please let us know if there are any other unclear notations.
>
> $\textbf{2. Please use citep{} for inline citations...}$
>
> We’ve modified the citation to \citep{} for indirect in-line citation.
>
> $\textbf{3. Most existing CL methods are unimodal,...}$
>
> As the reviewer mentioned and after thinking, we agree our writing was misleading. We updated the text accordingly in the $\textbf{second paragraph of introduction}$.
> Unimodal methods can be applied to multi-modal tasks in unimodal-wise, but they miss the cross-modal information which will be reflected in an inferior performance. Different from unimodal tasks which only consider the input of one modality, multi-modal tasks, vision-language tasks specifically, consider not only vision and language inputs but also the interactions between the two modalities. Simply adopting unimodal methods into vision-language tasks can barely preserve the feature interaction thus making the method sub-optimal. Our proposed method, instead, considers the modality interaction, based on the different combination of visual and text prompt key, and the experiment result shows the outstanding performance compared with other uni-modal baselines.
>
> $\textbf{4. All model parameters, except for the final classifier layer, are frozen during training to preserve the model’s generalizability...}$
>
> This is an insightful feedback. We add new analytic experiments about the performance after freezing different layers of the decoder of the transformer. The result is presented in table 6. Based on the table, we observe that there's a positive correlation between performance and layer freezed. Our hypothesis is that the prompt learnt for each task is specifically related to the model’s parameter at that moment. If too many parameters are updated through the training of the following tasks, when applying the same prompt for a task, the output feature will shift drastically due to the shift of the model's own parameters. Thus, freezing the whole transformer model gives us the best result.
>
> Besides, we acknowledge that many CL methods don’t freeze the model and fine-tune the whole model with the help of the continual learning algorithm. However, not all CL methods use this approach. Our method belongs to a relatively new category, prompt-based methods, introduced in 2021. The prompt-based methods are known for their efficiency as they freeze the whole model and only fine-tunes the prompts. These methods freeze the transformer layers to prevent catastrophic forgetting and use prompts to adapt to new tasks to improve the performance on each task. It is also important to note that previous prompt-based CL methods such as L2P, DualPrompt and S-Prompt freeze all of the transformer layers and still perform much better than methods that do not freeze the base model. Hence, we think whether training the whole model or not is not the key point in continual learning, as long as one can improve the performance and prevent catastrophic forgetting of a single model.
>
> In the case of CluMo, the reviewer raise a concern about our point: “by training only the classification layer, the model is forced to capture all task-specific knowledge through a single layer”. We want to clarify that the classification layer is not designed to be task-specific but to capture general VQA knowledge. We hypothesize that the task-specific knowledge is mainly obtained by the task-specific modality fusion prompts instead of classification layer. It is true that during training of different tasks, the single classification layer can learn task-specific information, but knowledge distillation constrains the parameter shift of classification to focus too much on any single task, which makes the classification layer contain general knowledge across all tasks.

---

> ### Author Response · Authors · 2024-11-04
> **Continued**
>
> $\textbf{5. By introducing an additional key pool to the learning algorithm, more trainable parameters are effectively added...}$
>
> Please note that showing the advantage of vision-language prompts over uni-modal prompts is a key component of our experiment and we have presented it in Table 2 as “Ablative Textual Key” and “Ablative Visual Key”. By default CluMo setting, we have 3 visual keys and 3 textual keys per task so 9 prompts in total to choose from. In “Ablative Textual Key” experiment, we eliminate the visual key so we have 9 textual keys to match to 9 prompts to choose. The prompt number remains the same in both cases. We observe that “Ablative Textual Key” is more than 2.5% worse than having multi-modal keys, which proves the importance of “multi-modal” keys and prompts. The result of “Ablative Visual Key” indicates the same thing. These experiments are also important to demonstrate that cross-modal interactions are informative.
>
> $\textbf{6. Do the other baseline methods also incorporate knowledge distillation?...}$
>
> Knowledge distillation is not included in other methods. We kindly remind the reviewer that the ablation study is presented in Table 2 “Ablative KD”. As it shows, without knowledge distillation, the accuracy and forgetting rate are still superior compared with other baselines, and it has the smallest performance drop compared to other ablation experiments, which shows knowledge distillation is not the “core” reason behind the supervisor performance of CluMo.
>
> $\textbf{7. Table 4 presents some puzzling results, as the number of keys appears to have no noticeable effect...}$
>
> We’ve added the experiment result for prompt size equal to 1 to Table 4. As we expected, when it only has 1 prompt, the accuracy drops significantly, which shows the effectiveness of having multiple prompts. Based on the results, our hypothesis of the stability of variant prompt size is that each prompt should keep the balance between “general” knowledge and “specific” knowledge to improve the result. Without sub-domain division, the prompt learns mostly the general knowledge across the task. Letting each prompt match a sub-domain of the task can improve the accuracy by focusing on sub-domain-specific knowledge, but further separating the task into smaller sub-domains may harm the learning of “general” knowledge, which prevents the improvement of the model’s performance or even harm the performance a little bit.
>
> $\textbf{8. In my opinion, this work lacks sufficient motivation...}$
>
> We believe that continual learning in VLM is extremely helpful in the real world applications due to the training efficiency. It is especially important in applications such as autonomous driving and medical fields where data is collected gradually and model updates can improve performance significantly. For example, SOTA autonomous driving systems are now using VLM to take both camera input and some pre-defined text for downstream tasks such as object detection, and they need to fine-tune the VLM model with their driving data collected from users over time. As the weather or road condition change, vehicle types or traffic signs are constantly changing in real world, it is necessary to frequently update these model to catch up. As the data is received in time sequence, eg. 100K data during Summer 2024, and another 100K data during Fall 2024, we cannot wait until Fall to fine-tune the model with all data at once. The model should be fine-tuned from time to time which is in line with the continual learning setting. In this case, in order to learn from the new data and prevent catastrophic forgetting of old data, continual learning methods are necessary and important.

---

> ### Author Response · Authors · 2024-11-04
>
> We really appreciate the reviewer’s suggestions for the paper, and we are now improving our paper’s content and clarity based on your advice. Please feel free to let us know any other questions or suggestions.

---

> > ### Comment · Reviewer_Aqr9 · 2024-11-11
> > **Reply to the comments**
> >
> > Thanks for the extensive rebuttal
> >
> > 2./3./6./8.  OK.
> >
> > 1. It's much better now, in my opinion. It could still be improved, but this is also a matter of taste.
> >
> > 4. I see that your approach aligns with other prompt-based methods. However, I'm concerned about Table 6. It suggests that as more layers are frozen, the degree of forgetting increases. This raises questions about whether your method truly prevents catastrophic forgetting or if the frozen layers alone are responsible. That said, this issue seems to affect some of the other baselines as well.
> >
> > 7. The extra results in Table 4 address my concern, but I still find the results puzzling. Having 2x2 -> 4 prompts is equivalent to 10x10->100. This implies that only 4 general knowledge prompts are sufficient, which feels very limited to me.

---

> > > ### Author Response · Authors · 2024-11-12
> > > **Reply**
> > >
> > > Dear Reviewer,
> > >
> > > Thank you for the fast reply and feedback!
> > >
> > > $\textbf{1}$
> > >
> > > Thank you for recognizing the improvement of the paper's writing and notation. We will keep polishing our draft paper and make it clearer and easier to follow. We also want to kindly ask the reviewer if there is any further specific advice for us so that we can update the draft?
> > >
> > > $\textbf{2}$
> > >
> > > We want to kindly correct the reviewer that in Table 6, we are associating the number of frozen layers with the accuracy of the model, instead of forgetting. As a result, we conclude that the more layers are frozen, the degree of $\textbf{accuracy}$ increases.
> > >
> > > The reviewer also mentions that frozen layers are responsible for preventing catastrophic forgetting. We don't deny that freezing layers can help with performance and prevent catastrophic forgetting. We conduct an experiment by fine-tuning the model on the CLOVE-scene dataset while keeping all the transformer layers frozen but classification layers. We get an accuracy of 40.49 with a forgetting rate of 21.76, better than non-frozen fine-tuning, which means the freezing layers can help.
> > >
> > > However, as the goal of continual learning is to achieve good performance for every task that comes in time sequence, we consider $\textbf{forgetting rate}$ as a "secondary" metric compared with $\textbf{accuracy}$. In Figure 6, we observe that all prompt-based methods have higher accuracy than non-prompt-based methods at the very beginning, and this kind of focus on accuracy, not strictly forgetting rate (although the forgetting rate is still outstanding), is also a way to achieve better performance for continual learning. On the other hand, we perform all the baseline experiments following their original settings. As the three prompt-based methods are all popular and have a great number of citations,  we consider the legitimacy of prompt-based methods in continual learning, and their way to freeze layers, to be reasonable and sound.
> > >
> > > $\textbf{3}$
> > >
> > > We've done more analysis of the prompt key size. We found that $\textbf{the prompt keys are not evenly selected by}$ $\textbf{the input feature.}$ This could happen with the K-means clustering algorithm that some of the cluster centers are selected more frequently, due to the distribution of data, the initialization of the clustering center, the ratio of cluster centers vs. the size of input data, etc. As we apply the mini-batch K-means clustering algorithm instead of the K-means clustering algorithm, such a phenomenon gets even worse.
> > >
> > > We observe that when we choose 3 x 3 -> 9 prompts, the top 6 prompts (by top N prompts we mean the N prompts that have the top frequency to be chosen) are selected by 99.41% of all input instances, while the rest of the 3 only occupy 0.59% of data. In the case of 10 x 10 -> 100 prompts, we observe that the top 5 prompts are selected by 44.53% of inputs, top 20 prompts are selected by 78.99% of inputs. As a result, even though we have 100 prompts, most of the prompts are barely chosen at all.
> > >
> > > Our design is to make the choice of prompt as diverse as possible, but such bias is still inevitable in reality. We also observe that in baseline $\textbf{L2P}$, even though the pool size is 20, only 2 of them are frequently chosen. As a result, although the distribution of prompt selection is not "the" optimal, when we compare the diversity and performance with other methods, we still consider our method effective and state-of-the-art.
> > >
> > > We hope these explanations can address your concern.

---

> > > > ### Author Response · Authors · 2024-11-29
> > > >
> > > > Dear Reviewer:
> > > >
> > > > We want to make more updates on the prompt size analytic experiment. With the suggestion of reviewer Axz6, we modified the logic of inference steps from task-aware to task-agnostic. As a result, we re-ran all the experiments of CluMo and the performance is still state-of-the-art.
> > > >
> > > > We also updated Table 5, which presents the result of different prompt sizes. We observe that the accuracy gap between different prompt sizes increases, which makes this hyper-parameter more meaningful. For example, the difference between 2x2 and 3x3 was 0.22 in the previous setting, and now the difference has become 0.74, which is much larger. We also observe that having 10x10 is the second-best accuracy among all and this can still be explained by the “general knowledge-specific knowledge balance” we mentioned in the previous rebuttal. Although the “general knowledge” of each task decays when the total number of prompts increases, the “specific knowledge” grows strong enough to cover the loss of “general knowledge.
> > > >
> > > > We hope this update can resolve your concern.
> > > >
> > > > Best,
> > > >
> > > > Our Team

---

> > > ### Author Response · Authors · 2024-12-05
> > > **Follow-Up with the Reviewer**
> > >
> > > Dear reviewer,
> > >
> > > Thank you for your time and being engaged in discussions. We were wondering whether you could check the latest version of our manuscript and see whether there is any remaining concerns.
> > >
> > > Thank you,
> > >
> > > Our Team

---

> > > ### Author Response · Authors · 2024-12-12
> > > **Follow-Up with the Reviewer**
> > >
> > > Dear reviewer,
> > >
> > > Thank you again for your time and effort and your engagement during the post-rebuttal period. We understand that reviewing is a volunteer-based service and really appreciate your feedback. Given that it is very close to the holiday season, we would appreciate you can read our responses after your comment, check the updated manuscript and let us know if you have any remaining concerns? If possible, we wanted to conclude the process before the holiday season.
> > >
> > >
> > > Best,
> > >
> > > Our team

---

### Review · Reviewer_Azx6 · 2024-10-15

**Summary Of Contributions:**

- The work proposes a prompt-learning technique towards VQA in a continual learning (CL) problem setting.
- The method assumes a domain incremental learning (DIL) setup where the input distribution (or domain) changes across tasks but the output classes remain the same across tasks.
- Technique:
    - The proposal is to assign an input image to a visual prompt key and input text to a textual prompt key. These prompt keys are learned during training via k-means clustering.
    - The visual and textual prompt keys are then combined to serve as a multi-modal prompt key using which prompts are retrieved for the input.
    - These prompts are pre-prepended to input before feeding to a multi-modal transformer (ALBEF).
    - Each task has a different prompt pool.
- VQA has been studied using prompt learning techniques in CL setups before using visual and language prompts. Prompt learning techniques (S-Prompt) have used clustering to build prompt pool before. The contribution here seems to be combining the learned unimodal prompt keys to build multi-modal prompt pools.
- The authors show performance improvements on CLOVE across the benchmark.

**Audience:**

No

**Broader Impact Concerns:**

Nothing to add.

**Claims And Evidence:**

No

**Requested Changes:**

- Please see weaknesses. I consider all listed weaknesses important to be addressed. I would weight W1>W2>W3=W4>W5.
- In addition, I have a few questions:
    1. It looks like there are separate prompt-pools for each task. This would mean separate visual keys, language keys and prompt-pool for each task. If so, how do you know which prompt-pool should be used during inference since you do not know the task identity in domain increment learning?
    2. It is not clear to me why performance on first task goes up after task 4 in Figure 6. What is the intuition here? Are there meaningful commonalities between task 1 and task (5,6)?
    3. Prior work [3] which introduced CLOVE specifies an upper bound performance measure based on offline training at 48 and 57 points for scene and function (Tab 2 in [3]) respectively. It looks like CluMo exceeds the upperbound for CLOVE-scene. Can you explain how this is possible?
    4. (optional: not necessary to be addressed) Since this technique uses combination of visual and language prompt keys, I’m curious how this method performs on novel skill-concept compositions seen during training similar to [2]. I wonder if the sub-domains captured during clustering in CluMo carries over well to such tests.
- Typos:
    - First line under Experiments should be “reproduced”.
    - Should be “backward” in Algorithm 2 pseudo-code.
    - 5.1: under CL Tasks: Should be “separate”.
    - 5.1: under Metrics for comparison: Should be “CLOVE” not GLOVE?
    - A.1: should be “DualPrompt”

**Strengths And Weaknesses:**

### Strengths:

- S1: The method is simple and easy to incorporate since most of the components already exists such as unimodal prompts, clustering. Incorporating prompt-combination based key can be seamlessly integrated to existing prompt-learning techniques in VQA.
- S2: The performance on CLOVE is clearly better across the benchmark relative to models used for comparison.
- S3: The manuscript is mostly easy to follow barring minor issues.

### Weakness:

- W1: I am mildly alarmed by the lack of citations and comparisons with existing VQA CL based techniques and even more so by the lack of comparisons with existing VQA CL techniques that employ prompt learning.
    - A non-exhaustive list of related works are referenced below. [1] and [3] in specific are prompt learning techniques looking at VQA in a CL based setting. [3] has been cited but it has been in the context of CLOVE dataset; the method from [3] should also be used to benchmark performance since they introduced the dataset and baseline method. On similar lines, [1] looks at the same problems tackled in this work on filling the gap created by using unimodal prompt keys. In order to ground this work in literature, performing comparisons with the relevant prior works can be helpful.
    - [2], [4-7] are other related techniques in VQA-CL that are not prompt learning based. Comparison should also be provided with non-prompt learning based techniques to empirically make clear the value provided by using prompt-learning for this problem.
- W2: There is a lack of dataset coverage in evaluation. The method has been tested only using CLOVE. Related works have used other datasets for comparison:
    - [1] provides CL-VQA benchmarks for evaluation based on two popular VQA datasets -  VQA2.0 and TDIUC.
    - [2] uses VQA v2 and NExT-QA.
    - [4] and [8] provides benchmarks.
    - I would recommend additional testing that can enable contextualizing CluMo in this space. If not, sufficient justification should be provided.
- W3: The problem is not adequately motivated in the manuscript. The manuscript mentions adapting VQA to CL is challenging but do not explain why this is required and do not provide citations to prior work in this area. While this might be common knowledge for researchers in this space, the manuscript needs to have stand-alone support for this. I would recommend the authors add details on why CL is needed for VQA as opposed to considering it a static problem, for instance, dealing with changing distributions of real world visual scenes, natural language, and increase of web-scale data from diverse sources. This must be augmented with citations of related work to ground the problem.
- W4: Clarity
    1. 4.1 under Prompt Pool: This section talks about retrieving a visual prompt key given an input image and pre-pending the prompt to the visual input. As far as I understand, this is explaining the general concept of prompt learning in visual domain but not necessarily the algorithm that is followed specifically. Placing this information under proposed method is a bit misleading in my opinion. In the later sections, the technique seems to be actually extracting the two unimodal keys, then combining them to retrieve a prompt which is prepended to the input i.e. visual prompt is not separately pre-pended to the image (equation 6). Feel free to correct me if I misunderstood anything. I request the authors to clarify or re-word it if my understanding is correct.
    2. This section also says “similarity score, such as cosine similarity”. It looks like what is being used is l2 distance in equation 5. Please be specific and correct as needed to maintain consistency.
    3. “V” should be defined in problem description (sec 3).
    4. Sec 5.3: I would expect “ablative textual key” to imply text based prompt keys are removed which I believe is the case here as well (?). The phrasing “For ablative text experiments, we change the size of textual prompt key to 9” is misleading however because I would expect there to be just visual prompt key here if text key was ablated?
    5. Nit: Figure 6 can use “CluMo” for legend instead of introducing new naming “DualCluster”.
- W5: The novelty is marginally as it seems to be specifically on combining unimodal prompt keys as far as I understand. Feel free to correct me if there is a misunderstanding on my part.  I do not weigh this weakness heavily in my judgment.

### References:

1. Decouple Before Interact: Multi-Modal Prompt Learning for Continual Visual Question Answering (ICCV 2023)
2. VQACL: A Novel Visual Question Answering Continual Learning Setting (CVPR 2023)
3. Symbolic replay: Scene graph as prompt for continual learning on vqa task (AAAI 2023)
4. CL-CrossVQA: A Continual Learning Benchmark for Cross-Domain Visual
Question Answering
5. Multi-Domain Lifelong Visual Question Answering via Self-Critical Distillation
6. Task Formulation Matters When Learning Continually: A Case Study in Visual Question Answering
7. Enhancing Continual Learning in Visual Question Answering with Modality-Aware Feature Distillation
8. CLiMB: A Continual Learning Benchmark for Vision-and-Language Tasks (NeurIPS 2022)

---

> ### Author Response · Authors · 2024-11-04
> **Rebuttal**
>
> Dear Reviewer,
>
> Thank you for your time and efforts. We are glad that you have identified significant areas of strength. We thank you for suggestions that helped improve our work. Please note that we have added new baselines, a new dataset, and a new backbone for more comparative experiments. We hope you engage in discussions and give us a second chance until we address all your concerns if we have not addressed all your concerns adequately. Our responses to the weaknesses you raised are below:
>
> $\textbf{Weakness}$
>
> $\textbf{1. I am mildly alarmed by the lack of citations and comparisons with existing VQA CL based techniques...}$
>
> The reviewer mentioned adding methods from reference [1]-[7] into the baseline. However:
>
> -After careful checking, we found that method [1], [4], [5], [6], are not open-source methods and their source codes are not provided to use. As a result, we cannot add them to the baseline due to the short time we had to prepare our response.
>
> -Method [3]: The reviewer mentioned the partial citation of method [3] and method [3] should be added into baseline. However, after consideration, we think it is not proper to add it into our baseline because all the methods from the experiments, including baselines and CluMo, are “add-on” methods with a pattern of “CL method + pre-trained backbone”, which means the backbone is replaceable, and we can take the CL method from the original backbone and fit it into other backbone. For example, EWC is a regularization-based method that introduces a new loss objective and is tested with the fully connected layer in the original paper. In our setting, we discard the fully connected layer and add the loss objective to “ALBEF”. We did this for all the baseline settings and we also added another backbone “BLIP” in Appendix A.1.2.  In case of method [3], the method is not an “add-on” method. The method is composed of a replay module “SRM” and a self-made transformer “UniVQA”. “UniVQA” is not a pre-trained, replaceable backbone that can be replaced with ALBEF or other model, it is part of continual learning method. In this case, if we adopt only “SRM” module to ALBEF, we only test the partial performance of method [3] which is not ideal. At the same time, if we compare the full version of method [3] with our baseline on CLOVE dataset, we observe that method [3]’s performance on CLOVE-scene with task order “abcdef” is 32.21, while the fine-tuning baseline from ours is even 34.03. The comparison doesn’t make any sense. Please also refer to the answer to the additional question 3 for further explanation.
>
> -As a result, we adopted method 2 and method 7 into our baseline and presented the result in table 1. We thank you for your suggestion that helped us to make our comparison more extensive.
>
> $\textbf{2. There is a lack of dataset coverage in evaluation...}$
>
> We agree with the reviewer that only testing the model on CLOVE dataset can be limiting. Although [4] offers a novel VQA-CL benchmark, but it is not open-resourced. [8] also proposes a continual learning baseline but it is not suitable for VQA as it also focuses on visual entailment and natural language natural reasoning. Thus, we formed a continual learning dataset which include 5 existing VQA datasets: COCOQA, OKVQA, GQA, VQAv2, TextQA. As these datasets are too large in terms of our computational resource (we only can use one A40 GPU each time) and time, we reduce the training dataset size to 20K and testing dataset for 3K for each dataset, which is the same setting as CLOVE dataset, by selecting the first 20K/3K data from the entire dataset. We present the result of the new comparison experiment in Appendix A.1, where CluMo is still the leading method compared with the baselines. We hope this addition addresses your concern.
>
> $\textbf{3. The problem is not adequately motivated in the manuscript...}$
>
> The application of CL-VQA is to adapt the model to the new image and text input from the real-world that comes in time-series and changes frequently. We add the motivation with citation to VQA methods which use continual learning in the $\textbf{Introduction}$ section with the blue mark. Please let us know if you would like to add more information in that section.
>
> $\textbf{4. Clarity}$
>
> $\textbf{(1)}$: We added new content in sec 4.1 to explicitly explain the difference between the general way to attach prompt and our way to attach prompt. The changes are in blue mark.
>
> $\textbf{(2)}$: We changed the “cosine similarity” to “L2 similarity” to match our experiment setting. The change is in sec 4.1 with blue mark.
>
> $\textbf{(3)}$: Given the suggestion of other reviewers, we changed and standardized the notation in section 3 and 4. V is replaced with other notations which are well-explained. The changed notations are marked in blue.
>
> $\textbf{(4)}$:  Your understanding of “ablative textual key” is correct. We acknowledge the misleadingness of our explanation. We modified the sentence in sec 5.3, marked in blue.

---

> ### Author Response · Authors · 2024-11-04
> **Continued**
>
> $\textbf{(5)}$:  We changed the name of “DualCluster” to “CluMo” in Figure 6.
>
> $\textbf{5. The novelty is marginally as it seems to be...}$
>
> We consider the novelty of CluMo beyond “simply combining unimodal prompt keys”, as we find a novel way to adopt the proper prompt guided by detailed-knowledge from both modality inputs which can explicitly take care of modality-interaction. In term of vision-language tasks such as VQA, the features should not only learn a good representation for stand-alone image or stand-alone text, but also need to learn the interaction between the two modalities to effectively align different modalities together, such as what CLIP model does with contrastive loss.
>
> Additionally, existing prompt-based CL method, such as L2P and DualPrompt, only consider image-modality input. If we apply them to VQA task, only the images is boosted with prompt while the text input remains the same. Method [1] proposes prompts for both image and text, although the code is not released, but the prompt attached to image only contains visual knowledge, and prompt attached to text only contains text knowledge. As a result, it can be regarded as two stand-alone prompts.  In our setting, we first divide the task’s image and text domain into sub-image domain 1,2,3, and sub-text domain 1,2,3. By applying a combination of sub-image domain and sub-text domain, the prompt is specified to learn the interaction between a subset of image and a subset of text, which introduces detailed and specific interaction knowledge into the prompt.  As a result, although our method might be “simple” to implement, we believe the logic behind that is innovative and novel, and the experiment has already proved the effectiveness of our method.
>
> $\textbf{Additional Questions}$
>
> $\textbf{1. It looks like there are separate prompt-pools for each task...}$
>
> The task_id is known during inference. The purpose of CluMo is to continually learn tasks which are from diverse domains, thus we claim it as “domain-incremental” in section 3. However, we agree with the reviewer’s comment “since you do not know the task identity in domain increment learning”, we modified our description in section 3 to task-incremental learning setting with diverse tasks in blue marks.
>
> $\textbf{2. It is not clear to me why performance on first task goes up...}$
>
> We hypothesize the phenomenon of performance increasing during task 5 and 6 is due to forward transfer. The dataset order of Figure 6 is $\textbf{ShoppingAndDining}$, $\textbf{WorkPlace}$, $\textbf{HomeOrHotel}$, $\textbf{Transportation}$, $\textbf{SportAndLeisure}$, $\textbf{Outdoors}$. As the scenario of $\textbf{WorkPlace}$, $\textbf{HomeOrHotel}$ and $\textbf{Transportation}$ are very different from $\textbf{ShoppingAndDining}$, the accuracy on $\textbf{ShoppingAndDining}$ drops significantly after learning those three tasks. In another word, the model is “damaged” regarding the $\textbf{ShoppingAndDinning}$ task. At the same time, dataset $\textbf{SportAndLeisure}$ and $\textbf{Outdoors}$ may contain data which are much more relevant to “ShoppingAndDining” compared with the rest of the three dataset.
> As a result, after learning $\textbf{SportAndLeisure}$ and $\textbf{Outdoors}$, the model, again, learns some knowledge related to $\textbf{ShoppingAndDining}$, which makes the performance increase a little bit, which is forward transfer. Meanwhile, as $\textbf{SportAndLeisure}$ and $\textbf{Outdoors}$ are just a little bit similar to $\textbf{ShoppingAndDining}$ the performance increases just a little bit instead of a lot.
> Please feel free to contact us if you have any questions.
>
> $\textbf{3. Prior work [3] which introduced CLOVE specifies an upper bound...}$
>
> The reason that CluMo beats the upper bound of performance in [3] is that we use a different transformer backbone. Method [3] uses self-made “UniVQA” model while we are using ALBEF. By only comparing fine-tuning between the two, in method [3] it achieves 27.53 on the CLOVE-scene with task order abcdef, while that in our paper is 34.03. It shows that ALBEF is better than “UniVQA”. Thus, CluMo’s performance is better than the offline accuracy in [3].
> This shows the importance of having the same backbone compared with the other baseline, which is why we changed all the baseline methods’ backbones to ALBEF. This also explains why we consider not to add [3] into baseline, as “UniVQA” is part of its proposed continual learning method and cannot be replaced by ALBEF.
>
> $\textbf{4. (optional: not necessary to be addressed) Since this technique uses combination...}$
>
> We think this is an interesting topic which may be tested in the future. However, due to the time limit and computational resource limit, we may not address this issue during the two-week period.

---

> > ### Comment · Reviewer_Azx6 · 2024-11-18
> >
> > Thank you to the authors for the rebuttal, however my concerns remain:
> >
> > - I acknowledge and even share the authors’ plight about some of the referenced techniques not being made public. At the same time, those techniques would still need to be cited as they address the exact problem this manuscript attempts to - VQA as a CL problem. The current version of the manuscript ignores a large body of directly relevant work. The version right now implies this is the first work to look at VQA-CL as a multi-modal problem which is untrue and can be misleading to potential readers. Citations can be provided along with qualifiers where appropriate to explain the methods are not public which prevents comparison etc.
> > - [6] seems to be publicly available here - https://github.com/MalvinaNikandrou/contvqa/tree/main
> > - Re ref [3]: “The comparison doesn’t make any sense” - Not sure if I agree with this. From what I can see, [3] is the only method (introduced as a benchmark on CLOVE) that uses CLOVE for comparison and is therefore the most direct comparison possible for the dataset and setting since CluMo and [3] tackles the same problem. Comparing with full method in [3] is okay and adding it to the table along with an asterisk explaining the nuances.
> >     - A big limiting factor to this right now is that the task order chosen in this work is different to that of [3] for both CLOVE-scene and CLOVE-function except for abcdef and oarlks in the two respective datasets. It is not clear to me why the order was swapped from the original work [3] introducing the dataset and benchmark.
> > - On dataset comparisons: The provided result increases my confidence. However, since it looks like a trend in prior works [1, 6] is to use VQA 2.0 dataset to curate a CL based dataset, using works like [6] could be more revealing.
> > - My biggest concern by far is the assumption of task identity being known. Domain incremental learning [9] is defined as the setting where domains can vary and task identity is not required to be known which is not true for CluMo as acknowledged in the rebuttal. In addition to updating the writing throughout the manuscript, there are far more implications. Most if not all techniques compared in the manuscript does not assume task identity (L2P, DualPrompt, S-Prompt) which muddies the water and even make this an unfair comparison. Moreover, the task identity being known at test time seems to be an impractical assumption for the problem addressed in this manuscript. If the setting here is task-incremental learning, where task identity is known, comparisons with techniques that employ different models based on the task at hand would be more appropriate along with grounding the motivation with this assumption.
> >
> >
> >     [9] Three scenarios for continual learning - https://arxiv.org/pdf/1904.07734

---

> > > ### Author Response · Authors · 2024-11-19
> > >
> > > Dear reviewer,
> > >
> > > Thank you for your continual engagement. We are grateful for your substantive comments and appreciate your time and constructive criticism. Given that your initial comment included points other than the ones in your last comment, our understanding is that other than the points you raised in your last comment, the remaining issues are addressed by our response. Please see our responses for the remaining concerns below.
> > >
> > > $\textbf{1. I acknowledge and even share...}$
> > >
> > > Part of these shortcomings is due to the page limit that we had. We think this concern is mostly about presentation and writing, and at the moment we are working on modifying the paper based on your suggestion. A request that we have is to please:
> > > - Let us know any uncited important works that we have missed. We are more than glad to include them somehow in the paper, e.g., in the Appendix if the page limit remains a challenge.
> > >
> > > - Please list the specific sentences in the paper that you think are overstating our contribution. This way, we can improve the tone and make them reflective of our contribution.
> > >
> > > $\textbf{2. [6] seems to be publicly available here...}$
> > >
> > > Thank you for providing us with the code link for [6]. However, we have two concerns about this paper:
> > >
> > > - This manuscript is unpublished and despite being available for two years, has not gained much attention. In our experience, comparisons usually are done with published works that have been peer-reviewed and verified unless an unpublished work has gained attention to warrant that it has been validated or there are no published works for comparison. None of these are true for this work.
> > >
> > > - More importantly, Paper [6] is an analytic paper that studies the effect of task setting on CL-VQA. It splits the VQAv2 dataset in three different ways based on “Diverse”, “Taxonomy”, and “Question”, and evaluates the performance of several existing continual learning methods on those datasets while using different transformers as the visual encoder to compare. As a result, it does not propose any new method that can be added as a baseline. We respectfully ask the reviewer to check the paper and its experimental setup.
> > >
> > > $\textbf{3. Re ref [3]: The comparison doesn’t make any sense...}$
> > >
> > > As the reviewer suggested, we will add the result from [3] into our comparison experiment as a baseline and explicitly point out the difference between this one and the others. We are currently running the experiment and this process will take some time to finish due to limitations of our computational resources.
> > >
> > > As for the difference in “task order” with respect to the experiment in [3], please note the authors explicitly mention that “we randomly sample 6 task orders from all possible permutations for evaluation”. This statement means that task order is not particularly an important task order. As a result, we can choose other task orders because the authors in [3] do not highlight the importance of task orders. Meanwhile, as we previously had not included [3] into our baseline, we didn’t strictly follow the task order randomly chosen by [3] paper and used the one that we initially used. The reason is that if we want to change the task order to match [3], then we need to redo all our experiments which will be quite time-consuming given our computational resources. (read the changes I did and make sure they are correct)
> > >
> > > $\textbf{4. On dataset comparisons: ...}$
> > >
> > > We are aware of the significance of the VQAv2 dataset in the field of visual question answering. Thus we have added it to our additional comparison experiments in A.1.1. The challenge for splitting VQAv2 into a continual learning sequence of tasks (like [1] and [6]) is that the author of [1] has only mentioned “we build continual learning benchmarks (denoted as CL-TDIUC and CL-VQA2.0) by dividing their images and questions into several disjoint hyper-categories, and then construct the benchmarks according to scenarios”. However, they have not explicitly mentioned how to split VQAv2, and the way they split VQAv2 is different from [6]. As a result, we are unable to follow their benchmark and make direct comparisons. We also raised concerns about [6] because it has been unpublished with marginal attention in the field for two years. Hence, we think it is rational to stick to our current datasets, CLOVE and self-made VQA datasets by the permutation of famous VQA datasets.

---

> > > > ### Author Response · Authors · 2024-11-19
> > > > **continued**
> > > >
> > > > $\textbf{5. My biggest concern by far is the assumption of task identity...}$
> > > >
> > > > Thank you for raising this concern which we have not given it a thorough thought.We think your concern is an important issue that needs to be addressed. Upon checking the literature after reading your comment, we noticed that [1] proposes an idea to infer the task identity during the inference in section 4.2.2 “Query and Match Strategy” which we think we can adopt to relax the need to know the next ID in our setting. In short, the idea that [1] introduces is to use a task-specific key and update it with a matching loss. During the inference time, the keys are used to compare with the input to find the best-matched task index. It would be great you read that section yourself and see it is applicable to our setting. Inspired by this approach, we plan to integrate the same (or similar) strategy to make it task-agnostic during inference. Our plan is to apply this change and regenerate the performance values only for “CluMo” in the domain-incremental version and update the result as fast as we can. However, we do recognize this is not the work that can be finished quickly in several days. We sincerely hope that the reviewer can give us the time to do so and address your concerns. We think two weeks should be sufficient to update our results. Please let us know if this change will address your important concern.

---

> > > > > ### Author Response · Authors · 2024-11-23
> > > > > **Domain Incremental Setting Preliminary Result**
> > > > >
> > > > > Dear Reviewer,
> > > > >
> > > > > Thank you for your patience and understanding, we are working on modifying CluMo to fit the domain incremental setting and we are getting some breakthroughs with preliminary results that we want to share with you.
> > > > >
> > > > > $\textbf{Task-agnostic Inference strategy}$
> > > > >
> > > > > After careful examination, we found that the “Query and Match Strategy” we mentioned in the previous comment does not fit our method. As a result, we came up with our own task-agnostic inference strategy that fits CluMo better.
> > > > >
> > > > > During inference, each test sample is sent to every task-specific pool and the best-matched prompt from every prompt pool will be gathered. For every prompt candidate, we calculate the $similarity_{total}$ = $similarity_{text}$ + $similarity_{image}$, where $similarity_{text}$ and $similarity_{image}$ are already calculated when selecting the best-matched prompt. We choose the prompt with the highest $similarity_{total}$ as the prompt and attach that one to the features.
> > > > >
> > > > > We adopted the above method and ran the experiment on CLOVE scene dataset, here are some preliminary results. We are still working on generating other results, which still need several days to finish.
> > > > >
> > > > > $\textbf{Preliminary Results}$
> > > > >
> > > > > | Task Order | Previous Acc. | Task-agnostic Acc. |
> > > > > |------------------|-----------------|-----------------|
> > > > > | abcedf     | 48.73     | 48.23     |
> > > > > | dbafec     | 48.80     | 47.52  |
> > > > > | bdcafe     | 48.83     | 48.18    |
> > > > > | acbefd     | 48.94    | 48.00     |
> > > > > | caefdb     | 48.26     | 47.97  |
> > > > > | bafedc     |  48.98     | 47.68     |
> > > > >
> > > > > From the table, we observe that by changing CluMo to the task-agnostic setting, there is a slight decrease of accuracy. However, the new results still outperform the baselines with significance. Thus, CluMo can still be state-of-the-art in the task-agnostic inference setting.

---

> > > > > > ### Author Response · Authors · 2024-11-29
> > > > > >
> > > > > > Following the previous reply:
> > > > > >
> > > > > > $\textbf{I acknowledge and even share...}$
> > > > > >
> > > > > > We have added the paper citations of [1],[2],[3],[5],[7], to the main paper in the second paragraph of $\textbf{Introduction}$ section. With the explanation, we offer the reader the sight of previous work and prevent the readers from mistakenly considering our work as the very first of the field.
> > > > > >
> > > > > > $\textbf{Re ref [3]: “The comparison doesn’t make any sense”}$
> > > > > >
> > > > > > We tried our best to implement [3] to add it to our baseline. However, we encountered the following technical issues:
> > > > > >
> > > > > > 1. We’ve spent a great amount of time trying to set up the environment for [3]. However, we stuck into the environment for “mmf”, which is a vision & language framework pipeline that required in order to run [3]. We have tried every single solution related to “Error Installing mmf” from
> > > > > >  https://github.com/facebookresearch/mmf/issues
> > > > > > and none of them work.
> > > > > >
> > > > > > 2. We also reached out to the author of [3] for more information on training and evaluating the model on the CLOVE dataset. Based on the author, they spend about 25 hours to train the model on a single task, eg. CLOVE scene abcdef, on a single A5000 GPU. Due to the computational limit, we can only access one A40 GPU at one time, which is 1.5 times slower than A5000. As a result, it will approximately take longer than 2 weeks to run [3] on 12 task orders listed in our baseline, even if the environment can be correctly configured.
> > > > > >
> > > > > > Moreover, we understand that as [3] is the origin of the CLOVE dataset, adding the method from [3] can improve the solidness of our experiment to some extent. However, as we already saw that the performance of [3], with its own designed backbone, is even worse than the fine-tuning baseline in our experiment, the performance of [3] is extremely unlikely to outperform CluMo with different task orders. As a result, with or without [3] in our baseline cannot make a significant difference from the perspective of the experiment result.
> > > > > >
> > > > > > Based on the issues above, we sincerely ask the reviewer to reconsider the request to add [3] to the baseline.
> > > > > >
> > > > > > $\textbf{My biggest concern by far is...}$
> > > > > >
> > > > > > We have finished all the experiments related to CluMo and updated the new result in the paper in Table 1-8. Overall, we observe a slight drop in accuracy compared with task-aware settings. However, the performance still outperformed all the other baselines with a meaningful margin. We also updated the description of the inference procedure in the last paragraph of sec 4.4. And we also updated Figure 6 with the new result.
> > > > > >
> > > > > > Best,
> > > > > >
> > > > > > Our Team

---

> > > ### Author Response · Authors · 2024-12-05
> > > **Follow-Up with the Reviewer**
> > >
> > > Dear reviewer,
> > >
> > > We are grateful for your feedback, particularly your concern about assuming the knowledge about the "task identity" which helped us to improve our work by relaxing it. We were wondering if you had time to check the updated manuscript and see if your remaining concerns are addressed?
> > >
> > > Thank you,
> > >
> > > Our team

---

> ### Author Response · Authors · 2024-11-04
>
> We really appreciate the reviewer’s suggestions for the paper, and we are now improving our paper’s content and clarity based on your advice. Please feel free to let us know any other questions or suggestions.

---

> > ### Author Response · Authors · 2024-11-12
> > **Follow-Up with the Reviewer**
> >
> > Dear Reviewer,
> >
> > We thank you for your time and the feedback you provided. We do understand that reviewing is a volunteer-based activity and your time is limited. However, it is crucial to benefit from this period and engage in discussions that hopefully can help addressing your concerns. We respectfully ask the reviewer to read our responses and let us know if any concern has not been addressed.
> >
> > Thank you again,
> >
> > Our team

---

> ### Author Response · Authors · 2024-12-12
> **Follow-Up with the Reviewer**
>
> Dear reviewer,
>
> We that your for your thorough and helpful review.  We are very satisfied that we received your feedback because your comments helped us to improve our work. We tried to address your concerns in the latest version of the paper. Given that it is very close to the holiday season, we would appreciate you can check the updated manuscript and let us know if your remaining concerns in your last comment are addressed? If possible, we wanted to conclude the process before the holiday season.
>
> Thank you again for your time and effort. We understand that reviewing is a volunteer-based service and you have a busy schedule.
>
> Best,
>
> Our team

---

### Review · Reviewer_npJ7 · 2024-10-21

**Summary Of Contributions:**

The paper addresses the critical issue of continual learning (CL) while mitigating catastrophic forgetting in visual question answering (VQA). Although CL has been extensively explored in unimodal settings (vision or text), the challenge of multimodal CL for VQA remains relatively underexplored, making this work particularly relevant. The authors propose the Cluster-based Modality Fusion Prompt (CluMo), introducing key-key pairs corresponding to vision and text prompts within a two-stage fine-tuning process. The method is rigorously evaluated on two variants of the CLOVE benchmark, demonstrating strong results.

**Audience:**

Yes

**Broader Impact Concerns:**

I couldn't find this section.

**Claims And Evidence:**

No

**Requested Changes:**

- The term 'prompt' was not clear to me throughout the paper. I assumed that you meant learnable soft-vectors as prompts. It was not clear to me until section “4.1 Preliminary.” This should have been clarified earlier, as this term is quite overloaded in the literature. A visual question, e.g., 'What is behind the door?' can itself be a prompt in the multimodal vision-language model literature.
- Mention the size of the ALBEF modules and the newly introduced modules in the proposed approach.
- Please provide a brief description of the baselines and a detailed description in the appendix. This should help the paper stand alone and allow for comparison with the proposed method. The related work covers this to some extent, but it would be better if it were properly organized.
- 'Figure 6: Accuracy on the first task after running the task sequence.' Improve the caption by clarifying the key takeaway.
- I'm curious about the computational complexity, given that the proposed approach is somewhat modular (for example, computing cluster similarity). How much faster/slower is the proposed method compared to the baseline ALBEF model? Discuss the inference and computational overhead of computing clusters.
- Provide experiments on another model. The authors may choose any model that fits their computational capacity, but a multimodal LLM model would make sense given the current experiments.
- In the related work section, please engage with more recent works in VQA. Recent foundation models like LLaVA and similar approaches are relevant since you are using a pre-trained ALBEF. One key shift is that VQA is now part of a larger pool of tasks trained in a single model, tackling tasks like OCR, document understanding, math reasoning, and more. This is a major trend change from the VQA v2-only models of a few years ago. How does this broader, multi-task setup relate to continual learning efforts?
- In the related work section, 'Prompt-Based Learning,' the writing feels somewhat loose. Are you suggesting that prompt learning is a powerful technique, and that your contribution lies within this domain? It would be helpful to provide a clearer thematic progression and position your work more explicitly within the context of existing research. This would enhance clarity and better highlight how your work advances the field.
- There’s a discussion on the benchmark dataset used, including statistics, in the appendix. It would be better to have a concise version of that in the main paper.
- I noticed that in CLOVE, there’s a notion of task difficulty as we continue learning; tasks should follow a curricular progression (from easier to more difficult tasks). However, the experiments in the table don’t seem to reflect that such a curriculum matters. I’m curious about the point of continual learning – perhaps one can train the model on the full dataset, as the community has recently been doing with LLaVA-line works. I’d like to know what the authors think about this.

**Strengths And Weaknesses:**

## Strengths

- I found the proposed clustering approach to be simple yet very effective, especially in how it improves the utilization of multimodal data. The motivation behind the clustering, building on unimodal methods, is well-justified, and the multimodal fusion technique is similarly straightforward but powerful for solving the task. Overall, the approach feels intuitive and delivers substantial gains while minimizing catastrophic forgetting, particularly compared to baseline methods. I also appreciated the comprehensive evaluation throughout the paper, which helps solidify the findings.
- The proposed method has advantages over rehearsal-based approaches, particularly in its memory efficiency, while still outperforming state-of-the-art prompt-based continual learning models. The significant reduction in catastrophic forgetting is very promising, which should make it practically relevant to the CL community.
- I particularly enjoyed the findings in section 5.2, along with the thoughtful discussion on the limitations of regularization-based methods versus the strengths of prompt-tuning approaches. The ablation study in section 5.3 is also well-executed, providing clear insights into how each module contributes to the overall performance of the model.

## Weaknesses
- My main concern is the generalizability of the approach. In VLMs (or MLLMs), we now have three flavors of models: 1) text-encoder, vision-encoder, multimodal fusion, and classification head, e.g., ALBEF (author’s setup); 2) vision-encoder, decoder-LLM with autoregressive generation (e.g., LLaVA); and 3) multimodal unified Transformer (e.g., Chameleon). The author has chosen ALBEF, but there’s no discussion of other lines of vision-language models at all. The choice of ALBEF may be justified due to its modular architecture, which aligns with the proposed approach and allows for easier integration of prompt-based fine-tuning. However, ALBEF is no longer the state-of-the-art vision-language model. Evaluating the method on more recent models, such as LLaVA, is important to understand whether the proposed solution works for other architectures, particularly those that use autoregressive generation. This comparison would also help the community better understand the VLM’s forgetting issues in light of recent progress and increase the value of the findings of the proposed approach.
- The second weakness is related to the first one. Only one base model, ALBEF, has been tested, despite the availability of many vision-language models. I think testing the method on more recent models would provide a more complete evaluation. I suggest the authors include an evaluation of another base model, such as LLaVA. The presented results are encouraging, and testing on another model would further strengthen the paper and broaden its applicability.
- The third concern is the arbitrary sizes of the prompt keys. While the authors have addressed this in the ablation section, it would be beneficial to provide more clarity earlier in the paper. How were the key sizes determined? Does the size of the keys affect the model’s performance? How large or small should they be? Please specify the total number of keys, their dimensions, and sizes in the experiments section. Although Table 4 presents interesting results suggesting that prompt key size doesn’t significantly impact performance, this seems counterintuitive. Wouldn’t a more expressive prompt key lead to better results, and a smaller one to worse performance? Highlighting these findings earlier and directing readers to the ablation section would enhance clarity.

[1] Liu, Haotian, et al. "Visual instruction tuning." Advances in neural information processing systems 36 (2024).

[2] Team, Chameleon. "Chameleon: Mixed-modal early-fusion foundation models." arXiv preprint arXiv:2405.09818 (2024).

---

> ### Author Response · Authors · 2024-11-04
> **Rebuttal**
>
> Dear Reviewer:
>
> Thank you for your time and efforts. We are glad that you have found our work effective and listed our strengths fairly. We thank you for the suggestions that helped improve our work. Please note that we have added new baselines, a new dataset, and a new backbone for further comparative experiments. We hope you engage in discussions and give us a second chance until we address all your concerns if we have not addressed all your concerns adequately. Our responses to the weaknesses you raised are below:
>
> $\textbf{Weakness}$
>
> $\textbf{1 and 2: Generalizability of the approach and diverse backbone}$
>
> We agree that showing the result with multiple backbones can further justify the performance of our proposed method. The reviewer mentioned using the LLaVA model as an additional backbone, which is the “visual encoder, LLM-decoder” architecture. However, as the size of the smallest LLaVA is about $\textbf{7B}$, compared to ALBEF’s size of about $\textbf{300M}$, it is completely beyond the computational resources that we have access to during a two week period. As a result, we choose another “visual encoder, LLM-decoder” model, BLIP, as our additional backbone. We adopt BLIP-base, which contains about $\textbf{200M}$ parameters yet is more recent and than ALBEF. We tested CluMo and all other baselines with BLIP as the backbone on the Clove scene dataset, and the results are presented in Table 8. We also added some analysis in section Appendix A.1.2, marked in blue. We hope the new additions can address your concern about adopting $\textbf{CluMo}$ to the backbone with architecture different than ALBEF and also more recent.
>
> $\textbf{3: Arbitrary size of prompt key and the effect of key size}$
>
> We specified the size and number of keys and prompts in section 5.1. To make a clearer comparison, we added the experiment results for the prompt size equal to 1 in Table 4, which means for each task, there’s no choice but to use the same prompt for all input instances. As we expected, when we only use 1 prompt, the accuracy drops significantly, which shows the effectiveness of having multiple prompts. Based on the results, our hypothesis of the stability of variant prompt size is that: each prompt should keep the balance between “general” knowledge and “specific” knowledge to improve the result. Without sub-domain division, the prompt learns mostly the general knowledge across the task. Letting each prompt match a sub-domain of the task can improve the accuracy by focusing on sub-domain-specific knowledge, but further separating the task into smaller sub-domains may harm the learning of “general” knowledge, which prevents the improvement of the model’s performance or even harm the performance a little bit.

---

> ### Author Response · Authors · 2024-11-04
> **Continued Rebuttal**
>
> $\textbf{Requested Changes}$
>
>
> $\textbf{1. The term 'prompt' was not clear to me throughout the paper...}$
>
>  We added the concept of prompt in Prompt-based learning in the $\textbf{related work}$ section $\textbf{Prompt-Learning}$ paragraph, the new content is marked in blue.
>
> $\textbf{2. Mention the size of the ALBEF modules and the newly introduced modules in the proposed approach.}$
>
> We added the size of the original model and the newly added model at the end of sec 4.2, marked in blue. Overall, for each new task that comes in, there are only 0.025% of new parameters added to the original model.
>
> $\textbf{3. Please provide a brief description of the baselines and a detailed description in the appendix...}$
>
> We added the explanation and how each method works in Appendix A.3, marked in blue.
>
> $\textbf{4. Figure 6: Accuracy on the first task after running the task sequence...}$
>
> We expanded the caption for Figure 6 with some explicit conclusions of analysis results.
>
> $\textbf{5. I'm curious about the computational complexity...}$
>
> Given that the model is trained on each task for 5 epochs in our experiment setting, the total time to train a task with CluMo is about 1600 seconds, among that the training of the cluster takes about 60 seconds in total.  While fine-tuning the ALBEF model on the same task is about 1450 seconds, which is about 10% slower than directly fine-tuning the model.
>
> $\textbf{6. Provide experiments on another model...}$
>
> We provided the experiment result of BLIP, which is presented in Table 8 in Appendix A.1.2.
>
> $\textbf{7. In the related work section, please engage with more recent works in VQA...}$
>
> We expanded the writing of $\textbf{Visual Question Answering}$ in $\textbf{Related Work}$ section by adding more SOTA methods and discussion.
>
> $\textbf{8. In the related work section, 'Prompt-Based Learning,' the writing feels somewhat loose...}$
>
> We mostly rewrote the prompt-based learning section so that it focuses on introducing what is “prompt” and how prompt is used in NLP and CV, in order to make it more explanatory and logical.
>
> $\textbf{9. There’s a discussion on the benchmark dataset used,..}$
>
> We have a concise introduction of CluMo dataset in section 5.1 “$\textbf{CL task}$” and also introduce the domain of sub-tasks of both CLOVE scene and CLOVE function in the first paragraph of section 5.2. We kindly ask the reviewer what extra information the reviewer thinks is necessary to be added to the main section.
>
> $\textbf{10. I noticed that in CLOVE, there’s a notion of task difficulty as we continue learning...}$
>
> We are not sure where the term “task difficulty” is mentioned in CLOVE. Could the reviewer provide the reference so we can provide our response? In terms of continual learning, as the reviewer asks, it is designed for real-world scenarios where one cannot obtain all the data at once, as the data comes or changes gradually over time. For example, when a company maintains a chat model and collects user data to frequently improve the model. In these cases, CL is helpful to update the model over time without the need to wait and collect a large amount of data and improve the model as more data becomes available.

---

> ### Author Response · Authors · 2024-11-04
>
> We really appreciate the reviewer’s suggestions for the paper, and we are now improving our paper’s content and clarity based on your advice. Please feel free to let us know for any other questions or suggestions.

---

> > ### Author Response · Authors · 2024-11-12
> > **Follow-Up with the Reviewer**
> >
> > Dear Reviewer,
> >
> > We thank you for your time and the feedback you provided. We do understand that reviewing is a volunteer-based activity and your time is limited. However, it is crucial to benefit from this period and engage in discussions that hopefully can help addressing your concerns. We respectfully ask the reviewer to read our responses and let us know if any concern has not been addressed.
> >
> > Thank you again,
> >
> > Our team

---

> > > ### Author Response · Authors · 2024-11-23
> > > **Follow-Up with the Reviewer**
> > >
> > > Dear Reviewer,
> > >
> > > We appreciate your time and the feedback you have shared. We believe it is important to make the most of the post-rebuttal period opportunity by engaging in discussions that can help address your concerns. We kindly request that you review our responses and inform us if any of your concerns remain unresolved.
> > >
> > > Thank you again,
> > >
> > > Our team

---

> ### Comment · Reviewer_npJ7 · 2024-11-26
> **Reply to comments**
>
> Thank you for the thorough rebuttal. The overall response looks good to me. In particular, I appreciate the experiments on BLIP.
>
> - W1. I liked the rationale and incorporation of BLIP. The result also looks very promising in Table 8. Plus, the nuanced finding on the prompt-based method should better inform the community on varying architecture.
> - W2. Thanks for addressing the concern regarding the prompt key size. Please consider adding the provided explanation in the appendix and refer to it in Section 5.1, which might help readers to consider hyperparameters choices.
>
> - RC1 to 4 looks good to me.
> - RC5. Please add a discussion in the appendix on the computational aspects of CluMo. The current rebuttal should be fine.
> - RC 6 to 8 looks good to me.
> - RC9. The clarification for section 5.1 is to mention the dataset statistics (total size for each dataset, give an example of the question and answer for the tasks. The reference to the appendix is just plain "Appendix", please give proper reference to the exact section/subsection.)
> - RC10. By task difficulty, I mean learning task splits assumably have easy to difficult order: "object" > "attribute" > "relationship".  For example, predicting a question about an object is easier than predicting a relationship-related question between two objects (training on ObjectRecognition split should make the learning of RelationReasoning easier)! Learning to recognize an object first and then learning "attribute" and then "relation" could create a curriculum as attribute is about an object and the relationship between two objects, requires understanding of the object themselves.
>
> I also followed some of the issues raised by the other reviewers and the response seemed reasonably convincing to me. I think the paper is in much better shape now!
>
> Paper organization suggestion: The added result on BLIP2 in the appendix seems a bit unusual. I suggest authors move them to the main paper and organize them as they seem fit.

---

> > ### Author Response · Authors · 2024-11-29
> >
> > Dear Reviewer,
> >
> > Thank you for your response and we are glad that the supplemental materials are convincing to you. With the following suggestions:
> >
> > $\textbf{W2}$: We’ve added more writing to explain the prompt key size effect in Appendix A.4, and linked it to Sec 5.1. One thing to notice, as we follow reviewer AXZ6's comment for task-agnostic setting during inference, we updated the results of CLUMO for all the experiments. In the experiment of prompt size, we now observe a larger gap between different prompt size which makes the choice of prompt size more meaningful. At the same time, the prompt size of 10x10 is still similar to 3x3 and we still consider the explanation previously valid to explain this.
> >
> > $\textbf{RC5}$: We’ve added the time complexity analysis paragraph to the Appendix A.3.
> >
> > $\textbf{RC9}$: We’ve made the reference in 5.1 more specified by adding the specific appendix section and table reference.
> >
> > $\textbf{RC10}$: Thank you for your explanation! We acknowledge that every task has its task difficulty and learning different tasks in a specific order can definitely improve the overall performance. This is also called curriculum learning, as the reviewer mentioned, which is similar to continual learning but not exactly the same. In continual learning, we cannot assume the order of tasks, as the tasks’ existence is not known unless we encounter them. In contrast, tasks in curriculum learning are known before training, and can be arranged manually to set the order. As the CLOVE paper focuses more on continual learning, we consider the task order that follows task difficulty as a “by-product” of their exploration. In the CLOVE paper, the author randomly chooses different task orders to perform the evaluation. By random choosing, the task order cannot be guaranteed every time, thus following task difficulty from easy to hard is not always possible.
> >
> > $\textbf{Paper Organization}$: We’ve followed the reviewer’s suggestion to move the BLIP experiment to the main sections. We put the BLIP experiment in Sec 5.4 along with the diagram and the table.
> >
> > Best,
> >
> > Our team

---

### Decision · Action_Editor_WMvA · 2024-12-10

**Recommendation:** Reject

**Comment:**

Overall, reviewers are borderline with two leaning accept and one leaning reject in their final recommendations. Multiple reviewers express concern or uncertainty about the experimental settings and relationship to prior work. The AE believes this work will be significantly stronger with another manuscript revision and recommends rejection at this time.

**Audience:**

The topic area of continual learning (CL) in VQA is likely of interest to some in the TMLR audience; however, appropriate context to existing CL-VQA work is necessary to accurately inform readers.

Reviewer recommendations are split regarding this criteria.

**Claims And Evidence:**

The manuscript has undergone significant revision during the rebuttal period to adjust the claims and experimental setting in response to reviewer concerns. However, the AE feels further work is necessary to fully support the paper's claims.

Among these issues was appropriately placing the proposed continual learning (CL) in VQA in appropriate context with prior CL work in the area (see reviewer Azx6's comments). This includes updating writing to contrast with prior CL-VQA methods and update baselines / experimental setting to align with prior work. While some progress was made, many additions were not possible during the rebuttal period due to limited time.

Reviewer recommendations are split regarding this criteria.

**Resubmission Of Major Revision:**

The authors may consider submitting a major revision at a later time.